



# Landscape scale remediation reduces concentrations of suspended sediment and associated nutrients in alluvial gullies of a Great Barrier Reef catchment: evidence from a novel intensive monitoring approach

Nicholas J.C. Doriean,[1,2] William W. Bennett,[1] John R. Spencer,[2] Alexandra Garzon-Garcia[3], Joanne M. Burton[3], Peter R. Teasdale,[4,5] David T. Welsh,[1] and Andrew P. Brooks[2,*]

[1] Environmental Futures Research Institute, School of Environment and Science, Griffith University, Southport, 4215, Queensland, Australia
[2.] Griffith Centre for Coastal Management, Griffith University, Southport, 4215, Queensland, Australia
[3] Department of Environment and Science, Queensland Government, Brisbane, 4102, Australia
[4] University of South Australia, UniSA STEM, Scarce Resources and Circular Economy (ScaRCE), SA, 5000, Australia.
[5] University of South Australia, Future Industries Institute, SA, 5000, Australia.

Correspondence to Andrew P. Brooks (andrew.brooks@griffith.edu.au)

**Abstract.** Gully erosion is a major source (~40%) of fine suspended sediment pollution to the Great Barrier Reef. Mitigating this source of erosion will have a lasting positive impact on the water quality of downstream rivers and the receiving marine environment. Here we conduct a preliminary evaluation of the ability of intensive landscape-scale gully remediation to reduce suspended sediment and associated nutrient export from a catchment draining to the Great Barrier Reef. A novel suspended sediment monitoring network, comprised of a suite of new and established automated monitoring methods capable of operating in remote environments, was used to evaluate the water quality of a remediated gully, a control gully and their respective catchments. Suspended sediment concentrations were ~80% lower at the remediated site compared to the control site, indicating the remediation works were successful in stabilising the erosion within the gully. Dissolved and particulate nutrient concentrations were also significantly lower at the remediated site, consistent with the decreased sediment concentrations. The novel combination of suspended sediment measurements from both the gully channels and overland flows in the surrounding gully catchments suggests that sediment and nutrients at the remediated site are likely sourced from erosion processes occurring within the catchment of the gully (at relatively low concentrations). In contrast, the primary source of suspended sediment and associated nutrients at the control site was erosion from within the gully itself. This study demonstrates the potential of landscape-scale remediation as an effective mitigation action for reducing suspended sediment and nutrient export from alluvial gullies. It also provides a useful case study for the monitoring effort required to appropriately assess the effectiveness of this type of erosion control.





## 1 Introduction

Water quality in the Great Barrier Reef (GBR) is negatively impacted by fluvially sourced pollutants; primarily
suspended sediment, dissolved and particulate nutrients, and agrochemicals (Bainbridge et al., 2018; Bartley et
al., 2014; Brodie et al., 2012; Fabricius 2005). Land use change, such as, mining, agriculture (grazing and
cropping) and urbanisation associated with European settlement in the region since the 1860s has increased the
output of fine sediment and nutrients from the catchments draining into GBR (Bartley et al., 2018; Kroon et al.,

2016). Catchment tracing studies have consistently identified sub-surface erosion processes, particularly from
stream banks and gullies, as the dominant source of fine sediment delivered to the GBR (Olley et al., 2013;
Wilkinson et al., 2015a). Gully erosion in particular has been identified as the largest single source of suspended
sediment, estimated to contribute more than 40% of all fluvially transported sediment entering the GBR
(McCloskey et al., 2017). Recent research suggests that these sediments, particularly from grazing lands, also act

as a source of bioavailable nitrogen (Garzon-Garcia et al., 2018a; Garzon-Garcia et al., 2018b).

Gullying occurs when unconsolidated soils and sediments become exposed and eroded by fast flowing storm
runoff (Brooks et al., 2018; Casalí et al., 2009). Gully erosion is a natural process, however, land use changes
have increased the rate of gully erosion and subsequent sediment export (Prosser et al., 1994; Shellberg et al.,
2016). The tropical climate of the GBR catchment region creates intense rainfall events (often > 40 mm h$^{-1}$) that

can rapidly erode tonnes of soil from an actively eroding gully during a single storm (Brooks et al., 2015; BOM,
2020).

There are various types of gullies present in the GBR catchment region (e.g., hillslope, colluvial, ephemeral, and
soft-rock badlands), however, alluvial gullies likely represent the largest source of sediment, accelerated by land
use change, to the GBR. This due to alluvial gullies consisting of mostly fine (<63 µm) dispersive and/or slacking

sediments and that they are located on the floodplains or terraces of river systems, thus, increasing the chance of
sediment transport to major rivers and marine receiving environments (Brooks et al., 2013; Brooks et al., 2016;
Brooks et al., 2019). These characteristics, coupled with the high connectivity of the gullies to river channel
networks, mean that a large proportion of the eroded fine sediment and associated nutrients from alluvial gullies
will be exported to coastal waters (Brooks et al., 2009; Brooks et al., 2018; Shellberg et al., 2013).

Gully remediation efforts in GBR catchments have typically focussed on smaller scale gullies (i.e., hillslope
gullies), with the application of low intensity erosion controls such as cattle exclusion fencing, revegetation, and
the manual installation of tree branch and/or geotextile fabric check dams (Bartley et al., 2017; Wilkinson et al.,
2015b; Wilkinson et al., 2013; Wilkinson et al., 2018). However, these strategies are not well-suited for stabilising
the much larger alluvial gullies that are present in many GBR catchments. Recent research suggests alluvial gullies

require the intervention of intensive landscape scale remedial efforts in-order to stem further erosion and reduce
sediment export (Brooks et al., 2016; Brooks et al., 2018; Carey et al., 2015; Howely et al., 2018). There are
several alluvial gully erosion mitigation projects currently underway in major GBR catchments (e.g., the
Normanby and Burdekin catchments), which are trialling various remedial works, including: large-scale
earthworks (i.e., reshaping of active gully head-scarps and sidewalls); rock chutes, including the application of

geotextile matting; rock-capping and mulching of potentially erodible soils; and the installation of bed control and
water velocity reducing measures (e.g., check dams). Stock exclusion and revegetation are also important
mitigation measures implemented in these gully remediation projects, often in concert with other treatments. The
overall aim of these remedial trials is to ascertain the control measures that are capable of permanently reducing


alluvial gully erosion and associated sediment, as well as particulate nutrient export (Brooks et al., 2016; Brooks

et al., 2018; GA, 2019).

Here we aim to assess the effectiveness of landscape-scale remediation in reducing suspended sediment and nutrient (particulate and dissolved, nitrogen and phosphorus) flow event concentrations associated with a large alluvial gully system in a GBR catchment. We apply a novel gully water-quality monitoring approach that utilises a suite of new and established autonomous suspended sediment sampling methods suitable for use in ephemerally

flowing systems in remote locations (Dorian et al., 2020). This new approach enables accurate measurement of suspended sediment as well as particulate and dissolved nutrient concentrations, while meeting the financial and operational requirements of a monitoring program situated in a remote location.


## 2 Methods

### 2.1 Study site

The study site is located on a cattle station in the Cape York Peninsula region of Queensland, Australia. There are several gullies that have formed in the alluvial floodplain and terrace of the Laura River (Figure 1). The tropical climate of the region is characterised by wet (October to April) and dry (May to September) seasons. Approximately 95% of the annual rainfall (regional mean annual rainfall is 936 mm) occurs during the wet season (Brooks et al., 2014a; BOM 2020). The study site topography is relatively flat with undulating gradients, surrounded by sandstone ranges. The alluvial sediments comprising the floodplain/terrace are derived from the Laura River catchment, which is dominated by the Ordovician Hodgkinson Formation meta-sediments, late Jurassic/early Cretaceous Gilbert River sandstones, and Quaternary/Neogene Maclean Basalts (Brooks et al., 2013; Brooks et al., 2014b) (Figure 1).

Two gullies were used to evaluate the effectiveness of the remediation works. The remediated gully is the larger of the two which encompasses several gully lobes that drain into a central channel. The gully treatment area is ~0.6 ha, with a catchment area of 13.7 ha. The active secondary incision of the control gully is ~0.2 ha while the gully catchment area is 3.3 ha. Both gullies are situated in highly-dispersible and slaking sodic alluvium. Prior to remediation, both gully catchments would have undergone similar erosion processes (i.e., scalding, sheet erosion, rilling in the gully catchment, tunnel erosion, head scarp mass-failure, and gully sidewall erosion within the incised part of the gully). Erosion rates derived from repeated airborne LiDAR between 2009 and 2015 (before remediation activity), indicate the control gully produced slightly more sediment (61 t$^{-1}$ ha$^{-1}$ yr$^{-1}$) compared to the remediated gully (50 t$^{-1}$ ha$^{-1}$ yr$^{-1}$), based on gully catchment area. Note, LiDAR does not account for the surface erosion generated from the catchment area of each gully, which would be expected to be comparable on an area normalised basis. Hence, the difference in specific yields between the treatment and control would be less than indicated by the LiDAR data alone (Brooks et al., 2016). Furthermore, comparison of particle size distribution (PSD) (Sect. 3.2.3) from readily erodible soil collected from the two gullies, prior to remediation activities, showed there was no significant difference between the two gullies. It is likely that these soils would have eroded into suspended sediment in a similar manner, thus, it is assumed that the suspended sediment concentration (SSC) and PSD from the two gullies would have been very similar, pending any significant differences in water velocity. The remediation of the larger gully complex was designed to halt the highly active erosion within the rapidly incising part of the gully and slow the scalding and sheet erosion processes within the broader gully catchment through destocking and the construction of contour berms (Brooks et al., 2018)

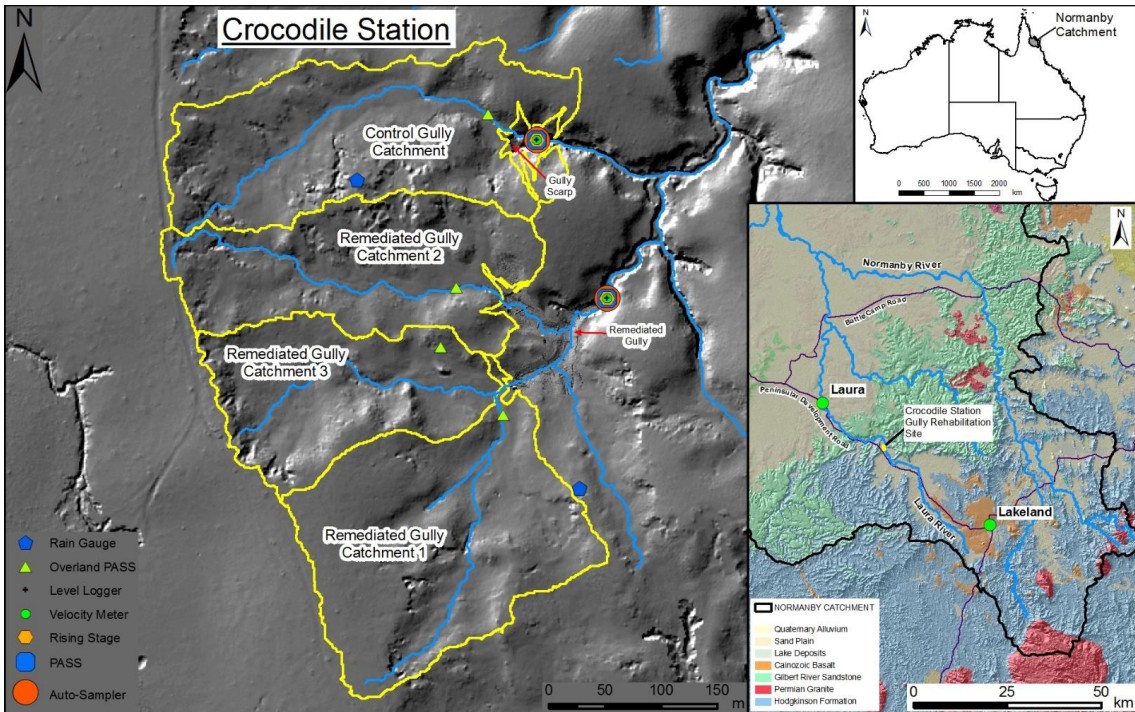

**Figure 1: Topographic map of the study site, including surface geology and gully locations. Source: (Geoscience Australia., 2019). PASS = Pumped active suspended sediment (PASS) sampler, Overland PASS = a PASS sampler used to sample water flowing overland (i.e., runoff), Rising stage = Single stage sampler (i.e., rising stage sampler).**

### 2.2 Gully remediation

The large actively eroding alluvial gully complex was remediated using various intensive, landscape scale gully erosion control earthworks during the 2016 dry season. The entire gully complex was regraded and compacted using heavy machinery. Gypsum was added during this process, to reduce soil dispersibility (Liu et al., 2017), and geofabric covering was applied over the former gully head scarp and held in place by a coarse sandstone surface capping. The rest of the gully complex was capped with locally sourced shale rock. Check dams were installed at regular intervals (approximately every 40 m) in the three major channels that replaced the original gully lobes (SI-1). After this, the entire gully complex was seeded with native vegetation and livestock were excluded from the gully and its surrounding catchment. No remedial efforts were applied to the control gully, other than the exclusion of livestock (SI-1) (Brooks et al., 2018). Time-lapse footage of the remedial works is available online at https://www.youtube.com/watch?v=dCbV1BggnKI (CYNRM, 2017).

### 2.3 Monitoring design

While gullies commonly share similar patterns of formation and erosion, there are many variables that need to be considered before implementing a monitoring plan to evaluate water quality within a gully system. Ideally, it is best to identify the factors that will have the greatest influence on gully water quality and monitor them prior to any remediation, in-order to establish a baseline of water quality conditions (i.e., a standard Before After Control



Impact (BACI) design). Any water quality monitoring assessment of a gully, particularly those being used to evaluate the effectiveness of remediation efforts, should provide a representative measure of the following parameters:

- Rainfall: the primary driver of continued gully erosion (Castillo et al., 2016).

- Soil: characterising basic soil physico-chemical parameters will aid in understanding the
140        transformation of soil into suspended sediment and how that may affect water quality (Brooks et al., 2016; Brooks et al., 2018).

- Water quality: it is recommended that at least two different means of water sample collection/measurement are used to ensure a representative measure of SSC and PSD. Entire flow events should also be monitored if possible (e.g., a time-integrated sample of an event is most
145        representative). If possible, samples should be collected from water flowing into the point of erosion (i.e., above the head scarp) and within the gully after the point of erosion (i.e., downstream of the head scarp) (Doriean et al., 2020).

In this instance the remediation project was required to implement the treatments and monitor the responses within a three-year timeframe, thus, a full BACI design was not possible. Instead, a control/impact design was used in
which remediation effectiveness was evaluated against a nearby comparable un-remediated control gully (SI-2). Three repeat airborne LiDAR surveys were collected over a six-year period, which enabled normalised baseline erosion rates to be calculated for the two sites, demonstrating the comparability of the treatment and control gullies (Brooks et al., 2016).

### 2.4 Monitoring methods

### 2.4.1 Hydrological and meteorological monitoring


Two rainfall gauges (Hydrological Services tipping bucket rain gauges - 0.2 mm/tip with Hobo data logger) were placed in the catchments of the remediated and control gullies (Figure 1). The rain gauges were programmed to provide a near continuous account of rainfall for the sampling period (2017/2018 and 2018/2019 wet seasons). Water level loggers (In-situ rugged troll 100®) were programmed to measure every two minutes and were secured
on the surface of a straight section of channel just downstream of each gully head (Figure 1). A barometric logger (In-situ barotroll®) was placed underneath the remediated gully rainfall gauge and set to record atmospheric pressure every 15 minutes.

### 2.4.2 Sample collection and monitoring

The original monitoring plan to evaluate the water quality conditions, focusing on suspended sediment, was
limited by funding and available measurement techniques, which resulted in only the outlets of both gullies being monitored for the first wet season (2017/2018). The successful modification of a recently established suspended sediment monitoring method, the pumped active suspended sediment (PASS) sampler, to operate in gullies (Doriean et al., 2020) allowed for the monitoring network to expand spatially and thus, enable monitoring of the time weighted average (TWA) SSC and PSD of sediment entering each gully from their respective catchments
during the 2018/2019 wet season.

Four different suspended sediment monitoring methods were used to collect water samples in the gullies: PASS samplers (Dorian et al., 2019), modified for gully deployments (Doriean et al., 2020); rising stage (RS) samplers


(Edwards et al., 1999); autosamplers (Edwards et al., 1999); and turbidity loggers (Gray et al., 2009 and Doriean et al., 2020). Several monitoring methods were used in this study to provide multiple lines of evidence to determine

the effectiveness of the remediation activities in reducing suspended sediment and nutrient export, as well as providing insight into the performance of the different monitoring methods. Each of the monitoring methods used in the control and remediated gullies were recently described and comprehensively evaluated by Doriean and co-workers (2020). The turbidity measurements recorded from the two gullies did not provide useful information for comparison of the gullies and there were few instances where turbidity measurements correlated with physically

collected samples. Therefore, turbidity measurement data collected from the gullies are not reported further here (see Doriean et al., 2020).

The TWA SSC and PSD of overland flows (i.e., catchment runoff) into the gullies was measured from samples collected using PASS samplers, configured to operate in ephemeral waterways (Doriean et al., 2020). The natural slope of the land flowing into the gullies had several depressions or low points that collected water as it flowed

over the land – PASS samplers were installed at these locations with the intake and float switch located 0.09 m above the ground (SI-2).

### 2.4.3 Soil sampling and analysis

Soil samples were collected as part of the design phase of the gully remediation project (Brooks et al., 2016; Brooks et al., 2018). Soil samples (1-2 kg) were collected from the face and walls of the gullies using a hand

trowel and auger. 21 and 9 samples were collected from the remediated and control gullies respectively, prior to the remediation activities. The soil samples were analysed for particle size distribution using the soil hydrometer method (ASTM standard method 152H) (Brooks et al., 2016).

### 2.5 Sample analysis and statistics

Collected samples were analysed for suspended sediment concentration (ASTM standard method D 3977-97) and

particle size distribution using laser diffraction spectroscopy (Malvern Mastersizer 3000, Malvern Instruments). Samples were screened using a 2 mm sieve prior to analysis to remove any large debris or detritus. Sediment used for particle size analysis was not chemically treated and was kept in suspension using mechanical (i.e., a baffled container with an impellor stirrer), reported in this study, or ultrasonic dispersion methods (Doriean et al., 2020). Nutrient analyses were conducted on a select group of samples. The samples were analysed for total and dissolved

organic carbon (5310 TOC and DOC 2017), and total and dissolved nitrogen and phosphorus (4500-Norg D and 4500-P B). Dissolved nutrient species (ammonium, oxidised nitrogen, and phosphate) were analysed segmented flow analysis methods: 4500-NH3, 4500-NO3, and 4500-P (APHA 2005; Garzon-Garcia, Bunn, et al., 2018; Garzon-Garcia et al., 2015). Due the remoteness of the field sites and sporadic nature of flow events, it was only possible to collect nutrient samples from the autosampler in a timely fashion on the 24th of January 2018 and the

6th of February 2019. Nutrient samples were not retrieved from the other instruments (Manual, RS, or PASS samplers) because the samplers contained samples from previous flow events, or the samples could not be collected and processed within the 48-hour timeframe. Consequently, the percentage of sand was likely underestimated in the samples, collected by the autosampler, which were analysed for nutrients (Doriean et al., 2020).





GraphPad-Prism® was used for statistical analysis of sample data following an evaluation for equality of group variances using Brown-Forsythe and Bartlett's tests before being analysed using paired t-tests to assess differences between sample groups (p = 0.05). The data was found to be normally distributed. Pearson's correlation analysis was also used to assess the relationship between SSC and nutrient concentrations.

**2.6 Data quality and uncertainty**

Throughout this study we attempt to acknowledge the uncertainty associated with the various monitoring techniques. A previous evaluation of the sample collection methods was used during this study to determine the approximate uncertainty associated with each method (Table 1) (Doriean et al., 2020). These uncertainties were accounted, for when interpreting data from the various methods, by taking into account measurement method error when comparing sampler data and data between the control and remediated gullies.


**Table 1. Uncertainties, of either SSC or PSD measurement, associated with suspended sediment monitoring methods used in alluvial gullies. Source: (Doriean et al., 2020).**

| Sampler type | Uncertainty (%) | | | |
|---|---|---|---|---|
| | TWA SSC* | PSD $d_{10}$ | PSD $d_{50}$ | PSD $d_{90}$ |
| **Autosampler** | 25 (± 10) | 10 | 25 | 45 |
| **RSS** | 20 (± 10) | 9 | 12 | 2 |
| **PASS sampler** | 9 (± 5) | 10 | 20 | 20 |

*TWA SSC = time weighted average SSC. RSS = rising stage sampler.*





## 3 Results and Discussion

Samples were collected from approximately half (5-6) of all flow events (> 0.2 m peak water level) recorded for the 2017/2018 wet season. Fewer events (3-4) were sampled during the 2018/2019 wet season due to two major backwater flooding events at the study site, caused by high water levels in the Laura River (see SI-3 for hydrographs of all sampled events). These flood events damaged equipment and contaminated samples with flood water. However, the flood events did not appear to affect the erosion mitigation structures of the remediated gully (SI-4). Despite the challenges of monitoring these remote systems and the unpredictable nature of flow events, sufficient samples were collected from a range of flow event types (i.e., intensity, length, time of year; SI-3) to meet the objectives of the study.

### 3.1 Rainfall and major hydrological events

Rainfall totals at the study site for the 2017/18 (920 mm) and 2018/19 (915 mm) wet seasons were not significantly different from the yearly average (943 ± 283 mm) of the permanent rain gauge operated by the Queensland Department of Natural Resources, Mines and Energy (DNRME), located at Coal Seam Creek, ~13 km from the study site. The on-site and DNRME rain gauges were in broad agreement ($R^2$=0.50; SI-5), although the variability in the relationship confirms that on-site rainfall gauges should always be deployed to achieve accurate rainfall intensity data. While there were many intense storms that resulted in flow events in the studied gullies, there were two major flood backwatering events that occurred in the 2018/19 wet season as a result of high-intensity rainfalls in the region surrounding the study site (SI-3). Review of historical DNRME stream gauge water level data of the Laura River, at Coal Seam Creek, showed that these backwatering events typically occurred with a ~3-year frequency over the 20-year dataset (DNRME, 2019).

### 3.2 Impact of remediation on suspended sediment characteristics

Soil characteristics and erosion estimates for the control and remediated gullies (prior to remediation), based on catchment size, area of readily erodible gully soil, and repeat Lidar aerial measurements suggest the control and remediated gullies likely had similar suspended sediment dynamics (Brooks et al., 2013; Brooks et al., 2016). The following sections describe how PSD, SSC, and most nutrient concentrations of samples collected from the remediated gully were significantly different/lower than the control gully for both wet seasons (2017/2018 and 2018/2019). A time series of all monitored flow events is included as supporting information (SI-3).

#### 3.2.1 Suspended sediment concentration

The remote location and challenging monitoring conditions typical of alluvial gullies meant that multiple suspended sediment sampling methods were used to ensure the most representative data were collected throughout both wet seasons (Dorien et al., 2020). Overall, the SSC range of samples collected by each method, from the outlet of the remediated gully were significantly lower compared to those collected from the outlet of the actively eroding, control gully (Table 2).

PASS sampler data were used to compare time-weighted average (TWA) SSC and other suspended sediment characteristics (i.e., PSD and SSC by sediment particle size class) of the remediated and control gullies because the method collected samples with the most representative PSD and TWA SSCs (Dorien et al., 2020), and





monitored the most flow events for both wet seasons (SI-3). The low temporal resolution of PASS sample data, theoretically, allows for the potential underestimation of SSC when very high SSCs are present at high flow rates for only short periods over the duration of a flow event (Doriean et al., 2019). However, comparable SSC data collected by manual flow proportional sampling, autosamplers, and RS sampler methods, which have high temporal resolution, corresponded well with the SSC range of the PASS samples from both gullies (Table 2), indicating that the PASS samples were representative of the measured events.

The median TWA SSC of PASS samples collected from the control gully (7123 mg L$^{-1}$) was five times higher than the median TWA SSC of samples collected from the remediated gully (1429 mg L$^{-1}$) (Table 3, Section 3.2.3), this, and statistical analysis, suggests there was significantly more sediment export due to erosion within the control gully than in the remediated gully. The TWA SSC of the catchment water flowing into the remediated (461-3556 mg L$^{-1}$) and control gully (485-2709mg L$^{-1}$) validate the assumption of similar contribution of suspended sediment from the two gully catchments during the monitoring period (see Table 4, Section 3.2.3). Comparison of remediated and control gully TWA SSC by sediment particle size class indicates the remedial works significantly reduced the concentration of suspended sand (96%), silt (76%), and clay (73%) (Figure 2). Bulk densities of the different sediment size fractions were very similar (~0.1 g L$^{-1}$ difference), and thus an average density was used to determine the different SSCs by size class (SI-6). The reduction in SSC across different sediment particle size classes indicates the remedial works are effectively reducing erosion and sediment export from the remediated gully. Continued monitoring of the remediated gully, for several more wet seasons, will be needed however, to determine the persistence of the sediment reductions associated with the gully remediation works.





**Table 2. Descriptive statistics of SSC samples collected from the control and remediated gullies, during the 2017/2018 and 2018/2019 wet seasons.**

| Sampling method | Remediated Gully | | | | Control Gully | | | |
|---|---|---|---|---|---|---|---|---|
| | AS | FP | RSS | PASS* | AS | FP | RSS | PASS* |
| Number of samples | 79 | 7 | 18 | 6 | 61 | 10 | 18 | 8 |
| Minimum (mg L$^{-1}$) | 350 | 364 | 378 | 1150 | 4146 | 3823 | 5675 | 5948 |
| 25% percentile (mg L$^{-1}$) | 827 | 421 | 906 | 1201 | 5055 | 4829 | 7874 | 6103 |
| Median (mg L$^{-1}$) | 1063 | 493 | 1502 | 1280 | 6180 | 5761 | 9177 | 7348 |
| 75% percentile (mg L$^{-1}$) | 1492 | 688 | 2736 | 2011 | 8162 | 6631 | 11278 | 8472 |
| Maximum (mg L$^{-1}$) | 3035 | 842 | 5278 | 2044 | 53086 | 8550 | 28696 | 14125 |
| Range (mg L$^{-1}$) | 2685 | 478 | 4900 | 895 | 48939 | 4728 | 23021 | 8177 |
| Mean (mg L$^{-1}$) | 1204 | 562 | 1860 | 1495 | 7773 | 5858 | 10560 | 7963 |
| Std. Deviation (mg L$^{-1}$) | 542 | 177 | 1275 | 411 | 6669 | 1331 | 5167 | 2670 |
| Std. Error of Mean | 61 | 67 | 300 | 168 | 854 | 421 | 1218 | 944 |
| Lower 95% CI of mean | 1083 | 398 | 1226 | 1064 | 6065 | 4906 | 7990 | 5730 |
| Upper 95% CI of mean | 1325 | 725 | 2494 | 1927 | 9481 | 6811 | 13129 | 10195 |
| Coefficient of variation | 45% | 31% | 69% | 28% | 86% | 23% | 49% | 34% |
| Sampler type | Are the control and remediated gullies significantly different? (α = value) | | | | | | | |
| AS | Yes (p < 0.0001) | | | | | | | |
| FP | Yes (p = 0.0001) | | | | | | | |
| RSS | Yes (p < 0.0001) | | | | | | | |
| PASS | Yes (p = 0.0007) | | | | | | | |

**AS = autosampler, FP = flow proportional sampling, RSS = rising stage sampler, PASS = PASS sampler. * = PASS samples represent the time weighted average suspended sediment concentration for the time the sampler was deployed.**

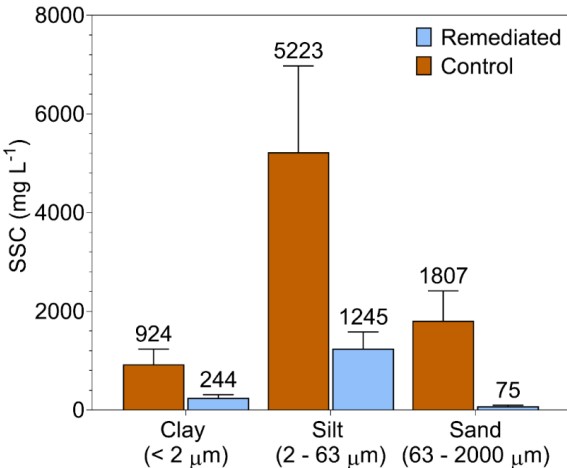

**Figure 2. Median SSC by sediment size class for PASS samples collected from the control (brown) and remediated (blue) gullies during the 2017/2018 and 2018/2019 wet seasons. Error bars represent sample standard deviation.**
**Autosampler and RS sampler SSC by PSD are included in SI-7.**



### 3.2.2 Relationship between SSC and flow

There is currently insufficient water discharge data to accurately estimate the sediment loads of the two gullies monitored in this study. The unstable nature of gully banks and bed features means the channel cross-section can change dramatically during a single event, thus obtaining an accurate measurement of the gully channel cross

section over a wet-season is rarely feasible. As a result, the use of a discharge related rating curve based on a single measure of channel cross-section will have high uncertainty (Malmon et al., 2007). Furthermore, manual measurements of water velocity can be very dangerous to perform, due to the risk of rapid water level rise (e.g., the control and remediated gullies can often encounter water level changes of 0.5 m in under 5 minutes) and the potential of bank collapse in the control gully. Automated methods for determining velocity or discharge (e.g.,

acoustic doppler velocimeters/acoustic doppler current profilers) offer an alternative to manual measurements, however, these methods are expensive and are limited to waters where SSC is typically less than 15000 mg L$^{-1}$, without additional site-specific calibration (Sottolichio et al., 2011). For these reasons it takes considerable time and effort to collect sufficient data to accurately determine gully discharge and, therefore, sediment load. Once an adequate amount of gully water discharge data are collected, sediment load estimates for the remediated and

control gullies will be calculated and published.

In the absence of water velocity data, comparison of water levels (and thus shear stress), likely to show similar trends to velocity, and SSC show that there was no obvious relationship for the control gully. However, SSC trends in the remediated gully, particularly in the 18/19 wet season, may be linked to water level, likely as a function of velocity (Figure 3) (SI-3). Additional flow event data, including water velocity measurements are

needed to confirm this.

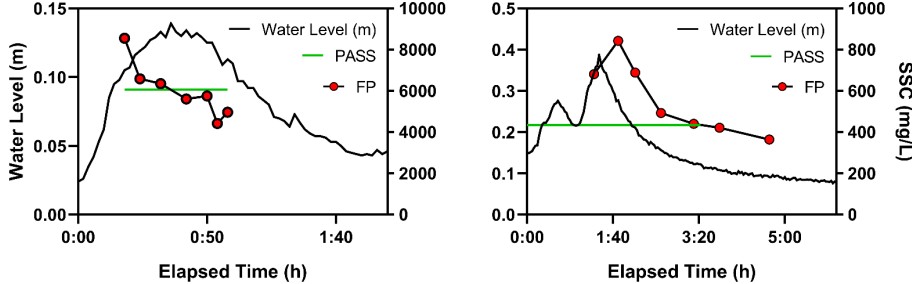

**Figure 3. Relationship between SSC and stream height for single flow events in the control (left panel, flow event B) and remediated (right panel, flow event F) gullies, that occurred during the 2018/2019 wet season (SI-3). Water level (black line), PASS TWA SSC (green line), and flow proportional (FP) sampling (red circles with black line).**





The SSC of samples collected from the control gully, using RS samplers and autosamplers, suggest there is a general decreasing trend in SSC following the initiation of flow ($R^2 = 0.61$), regardless of changes in flow event length or stage height (SI-3) (Figure 4). This trend is likely the result of instream processes, such as the rapid mobilisation of readily erodible soil from the gully and deposited fine sediment from previous flow events contributing to a high initial SSC followed by a steady decrease in SSC to an equilibrium between the scouring

of erodible gully soil source material and the transport capacity of the water flowing through the gully (Malmon et al., 2007). These processes have been observed in other ephemeral waterways and may be an inherent feature of these systems (Dunkerley et al., 1999; Malmon et al., 2002). In contrast, there was no relationship ($R^2 < 0.01$) between SSC and time after the initiation of flow in the remediated gully (Figure 4). The SSC trend in the remediated gully is no longer symptomatic of an actively eroding system, rather, it is a relationship similar to that

of streams transporting sediment sourced from the catchment (Doriean et al. 2019; Nistor and Church 2005). This suggests gully erosion is no longer the dominant sediment source and the gully may now be a conduit for suspended sediment sources from erosion processes occurring in the catchments.

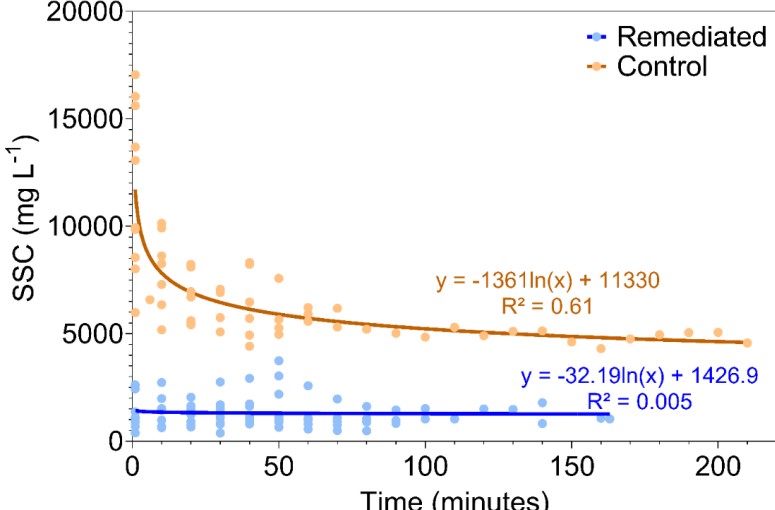

**Figure 4. Relationship between time after initiation of flow and SSC of samples collected from the control (brown) and**

**remediated (blue) gullies using autosamplers and RS samplers during the 2017/2018 and 2018/2019 wet seasons. Trend lines represent logarithmic regression models.**



### 3.2.3 Particle size distribution

The PSD of erodible soil collected from both the control and remediated gullies, prior to remediation, were not significantly different (SI-8) (Figure 5). For both gullies, ~45% of readily erodible soil from the gully head scarp
was comprised of sand, with the remainder being silt (~35%) and clay (~20%) (Figure 5). The near identical PSD characteristics of the readily erodible soil from both gullies is consistent with their proximity and indicates that the control gully provides an appropriate comparison to evaluate the effectiveness of remedial works at the remediated gully.

Suspended sediment samples from the control gully, collected using a PASS sampler, demonstrate the alteration
in PSD of the gully soil when it becomes suspended under flow, mixed with sediment from the catchment and selectively transported downstream (Figure 6). This change in PSD is expected because the sediment particles will distribute in the water column based on their physical and chemical characteristics, such as shape, size, mass, and affinity to flocculate into composite particles (Vercruysse et al., 2017; Walling et al., 2016). Hence, lighter and finer particles (clay and silt) were dominant in the suspended sediment samples. The bulk of the sand in the
eroded gully soil is likely transported as bed load, with the proportion in the suspended fraction dependant on periods of high flow-velocity (Horowitz, 2008). The presence of large deposits of sand within the control gully channel bed supports this interpretation (SI-9).

Comparison of the average PSD of suspended sediment samples collected from the remediated and control gullies show that silt and clay were dominant in both, however, sand was almost completely absent (<6%) in the
remediated gully samples (Figure 7). There was no visual evidence of bedload sediment (i.e., sand) settling in the remediated gully channel, rather, these coarser sediment particles (>63 µm) appeared to be trapped behind flow reduction structures (i.e., check dams) (SI-9). Comparison of suspended sediment PSD characteristics (10th (d10), 50th (d50), and 90th (d90) percentiles) of PASS samples collected from the control and remediated gullies show that the suspended sediment from the remediated gully (d50 of 5.84 µm) was significantly finer than that of the
control gully (d50 of 10.8 µm) (Table 3).

The suspended sediment PSD characteristics of control gully catchment PASS samples was notably different to the gully outlet PASS samples (Table 3). This indicates the contribution of slightly coarser suspended sediment from gully erosion (d50 10.8 µm) is greater than the suspended sediment contribution of the catchment (d50 4.29 µm) in the control gully. In contrast, the PSD characteristics of suspended sediment samples collected from the
outlet of the remediated gully (d50 of 5.84 µm) and samples collected from Catchments 2 (d50 of 5.52 µm) and 3 (d50 of 5.06 µm) of the three catchment areas draining into the gully were very similar (Table 3) (Figure 8). This suggests there is a notable contribution of sediment entering both gullies from their respective catchments. The lack of similarity in suspended sediment PSD characteristics between the remediated and control gullies outlets, and similarity in the PSD of the remediated gully and its catchments, indicates gully subsoil (i.e., sand
and coarse silt) is no longer a significant source of the suspended sediment flowing from the remediated gully. It also indicates that the dominant PSD component of fine suspended sediment (i.e., clay and silt) in the remediated gully is now primarily sourced from the gully catchments.





**Table 3. Time-weighted average suspended sediment concentration and particle size distribution data of samples**
**collected, using PASS samplers, from the remediated and control gullies during the 2017/2018 and 2018/2019 wet seasons. Note catchment samples (n=2 per sampling location) were only collected during the 2018/2019 wet season.**

| *Sampling location* | *TWA SSC (mg L$^{-1}$)* | *PSD (µm)* | | |
|---|---|---|---|---|
| | | *$d_{10}$* | *$d_{50}$* | *$d_{90}$* |
| Control gully | 7123 (± 2670) | 1.79 | 10.8 | 175 |
| Control catchment | 485-2709 | 1.04 | 4.29 | 26 |
| Remediated gully | 1429 (± 419) | 1.40 | 5.84 | 27 |
| Remediated catchment 1 | 337-563 | 1.71 | 8.11 | 36 |
| Remediated catchment 2 | 461-1517 | 1.27 | 5.52 | 30 |
| Remediated catchment 3 | 808-3556 | 1.27 | 5.06 | 24 |

**Please note, each catchment PASS sample TWA SSC represents the average SSC of several flow events.**

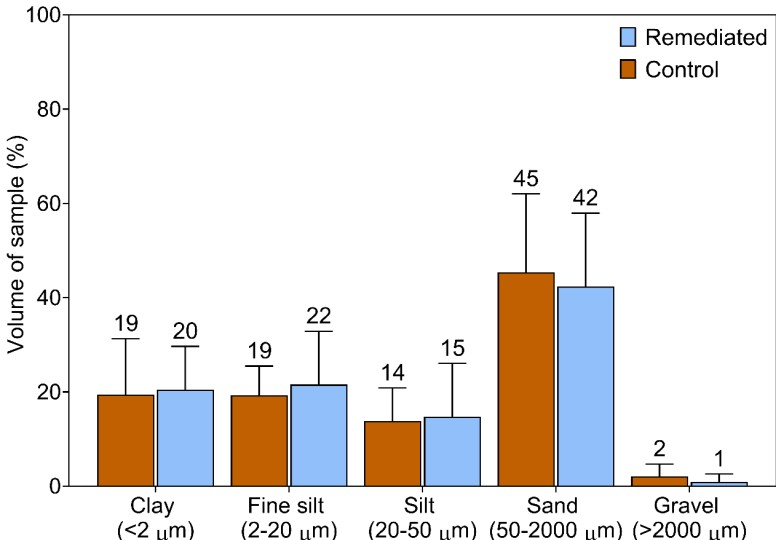

**Figure 5. Average PSDs, by size class, of soil collected from the control (brown) and remediated (blue) gullies, prior to**
**remedial works. Error bars represent the standard deviation of each class. Control n=4 and remediated n=14.**

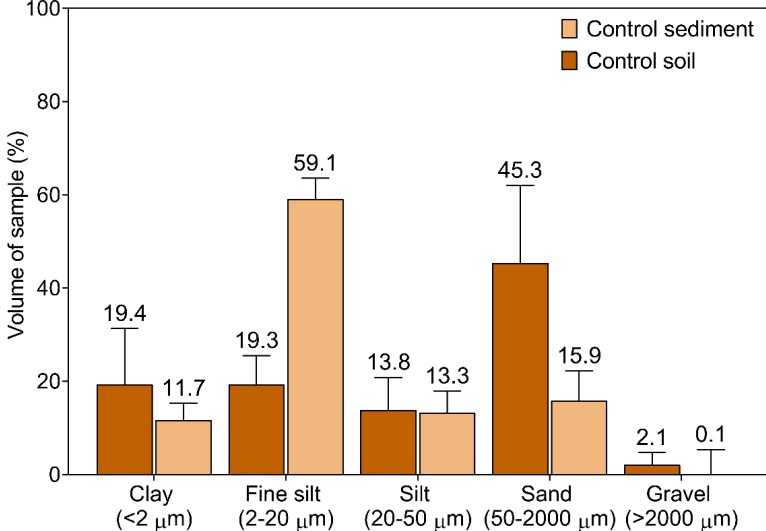

**Figure 6. Control gully soil (brown, n=4) and control gully suspended sediment (light brown, n=6) PSD by size class.**
**Error bars represent error as standard deviation for the soil and sediment PSDs respectively.**

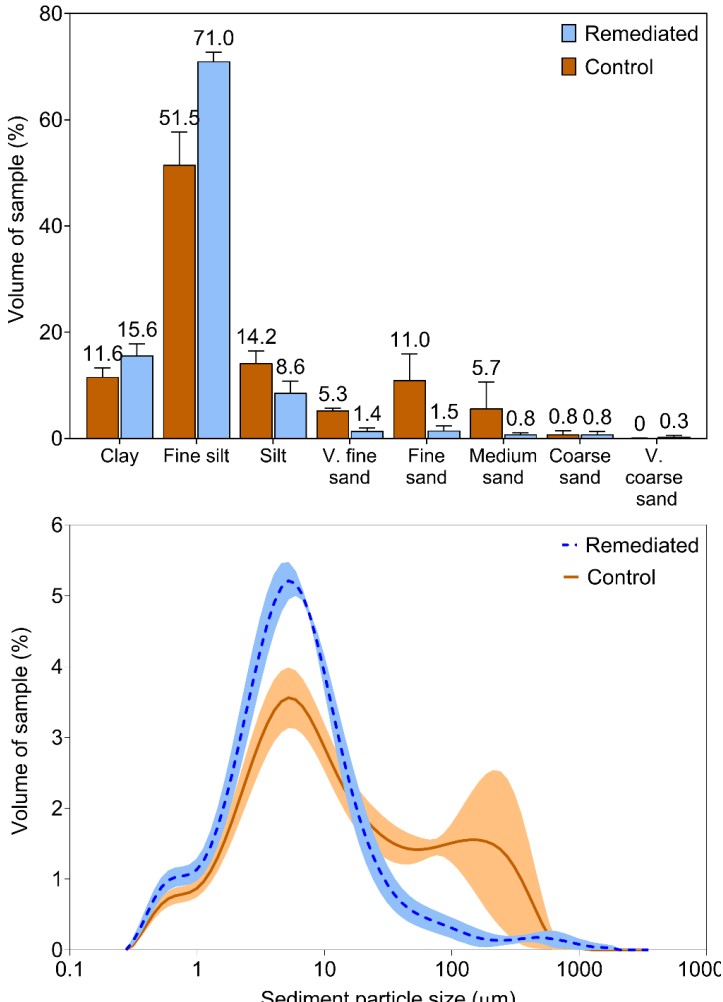

**Figure 7. Average suspended sediment PSD by sediment size class (left panel) and plotted by frequency (right panel) for PASS samples collected from the control (brown) and remediated (blue) gullies during the 2018/2019 and 2018/2019**
**wet seasons. Error bars (left panel) and shading (right panel) indicate error as standard deviation. Clay = <2 μm, Fine silt = 2-20 μm, Silt = 20-63 μm, very fine sand = 63-100 μm, fine sand = 100-250 μm, medium sand = 250-500 μm, coarse sand = 500-1000 μm, very coarse sand = 1000-2000 μm.**

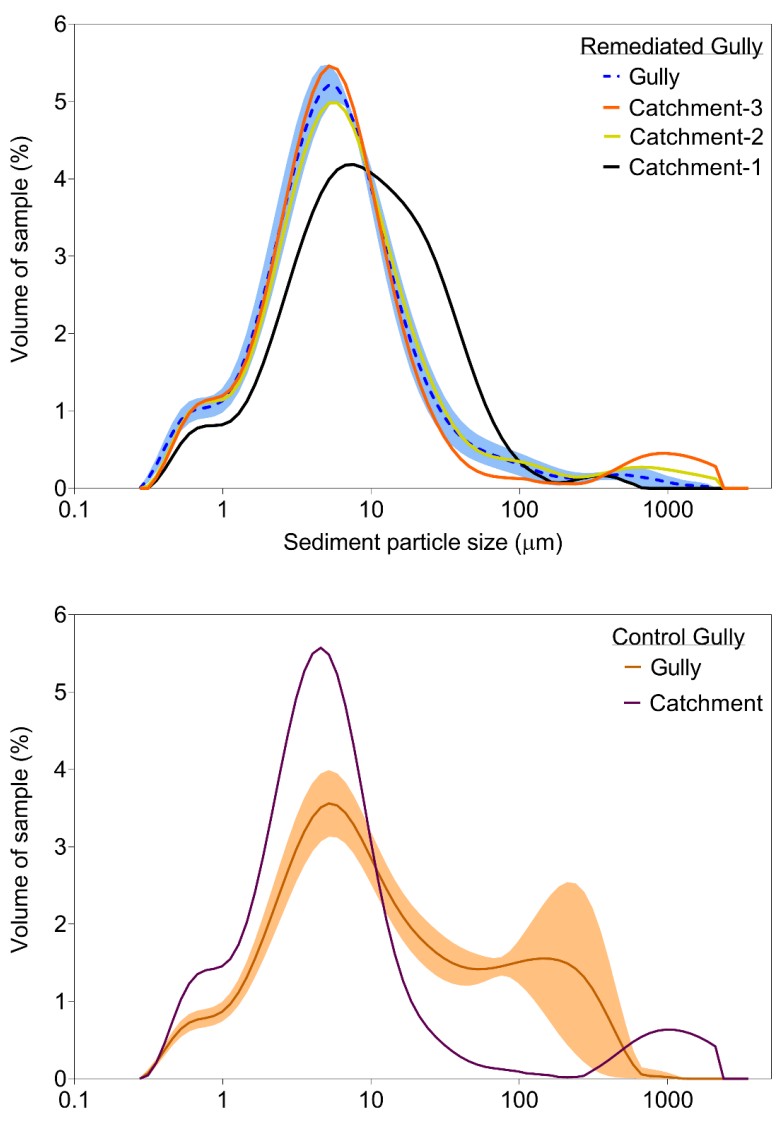

**Figure 8. Average PSDs of PASS samples collected from the remediated gully (blue) and catchments (orange, yellow, and black) and control gully (brown) and catchment (purple) suspended sediment PSD frequency plots, during the 2018/2019 wet season. Shading around gully PSDs represents error as standard deviation.**






### 3.3 Particulate and dissolved nutrients

Three opportunities occurred during the study period (24/01/2018, 15/12/18, and 05/02/2019) where samples were able to be retrieved from the remote sampling site within a time frame that allowed them to be processed (i.e.,
refrigerated samples filtered and frozen within 48 hours of collection) and analysed. The hydrographs and SSC trends of these events indicate they are representative of the flow events observed in the two gullies (SI-3). Note, the SSC of these samples were likely underestimated by ~15% because they were analysed using the total suspended solids (TSS) analysis method rather than the SSC method (Gray et al., 2000).

The bulk of total organic carbon and nutrient (nitrogen and phosphorus) concentrations, for both gullies, consisted
of particulate fractions (Figure 9). Organic carbon and nutrient concentrations of samples collected from the remediated gully were significantly lower than control gully samples for both dissolved and particulate fractions, except for dissolved organic carbon and nitrogen during the 2018/2019 wet season (Table 4; Figure 9).

Dissolved nutrients are influenced by numerous biogeochemical processes that occur in the catchment and the gully, with some of these processes occurring rapidly (i.e., instantly or within several minutes) and significantly
altering nutrient chemical speciation (Garzon-Garcia et al., 2016; Garzon-Garcia et al., 2015; Lloyd et al., 2019). We do not currently have sufficient information to investigate the effect these processes have on dissolved nutrient trends occurring in the gullies and their catchments, thus, our interpretation of this data will be limited. However, particulate nutrients and carbon are more stable, taking days or weeks to undergo large changes due to biogeochemical processes, that is after initial leaching of soluble components has already occurred (Garzon-
Garcia et al., 2018a; Waterhouse, et al., 2018). Therefore, we can assume that the particulate nutrients are relatively stable and representative of their source when sampled from the gully outlet.

For the samples collected during flow events on the 23rd of January 2018, the SSC and particulate nutrient concentrations showed a significant correlation in the control gully (r = 0.68 to 0.78; p < 0.01), whereas there was no significant correlation (r = 0.23 to 0.48; p > 0.05) between SSC and particulate nutrient concentrations in the
remediated gully (Figure 10; SI-10). The strong positive relationship between SSC and nutrient concentrations in the control gully supports the hypothesis that erosion processes within the gully are acting as the dominant source of suspended sediment and particulate nutrients. In contrast, the poor relationship between SSC and nutrient concentrations in the remediated gully is likely due to the much lower rates of gully erosion at this site, which limits the range of SSCs over which the relationship can be evaluated. It may also indicate multiple sources (i.e.,
sediment and detritus inputs from the catchment) are contributing to particulate nutrient export. The remediated gully suspended sediment had a significantly higher nutrient proportion by mass than that from the control gully (SI-11), consistent with the higher proportion of fine suspended sediments observed in the remediated gully, as a result of reduced subsoil erosion effects (Figure 7) (Horowitz 2008). Reliably differentiating fine suspended sediment and associated nutrients sourced from either the catchment or the gully itself is challenging without
dedicated sediment tracing data (e.g. stable or radioisotopes, biomarkers), and/or a distributed network of event samplers within the catchment. However, our PSD data is consistent with a dominant catchment source of suspended sediment and particulate nutrient sources in the remediated gully. Whereas, the significant relationships between SSC and particulate nutrients in the control gully demonstrates that eroding subsoil was a major source of particulate nutrients in the control gully. Future work should seek to investigate the specific sources of
suspended sediment and associated nutrients at the study sites.

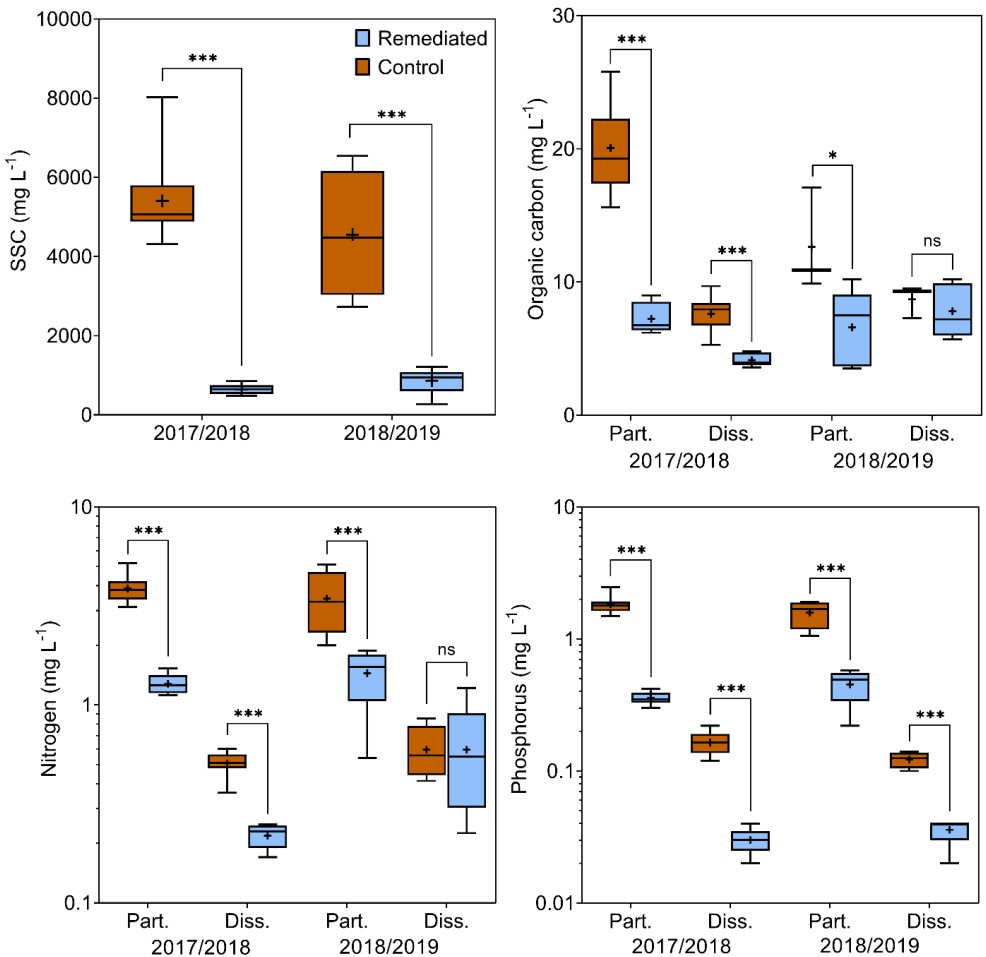

**Figure 9. SSC and nutrient concentrations of samples collected during flow events in the 2017/2018 and 2018/2019 wet seasons. Note, the 2017/2018 data represents a single flow event and the 2018/2019 data represent multiple flow events. Box plots represent the minimum, maximum, 25th and 75th percentiles, median (horizontal line in box), and mean (cross). Brackets represent the results of paired t-tests, where p < 0.001 (\*\*\*), p < 0.01 (\*\*), p < 0.05 (\*), or p > 0.05 (ns).**




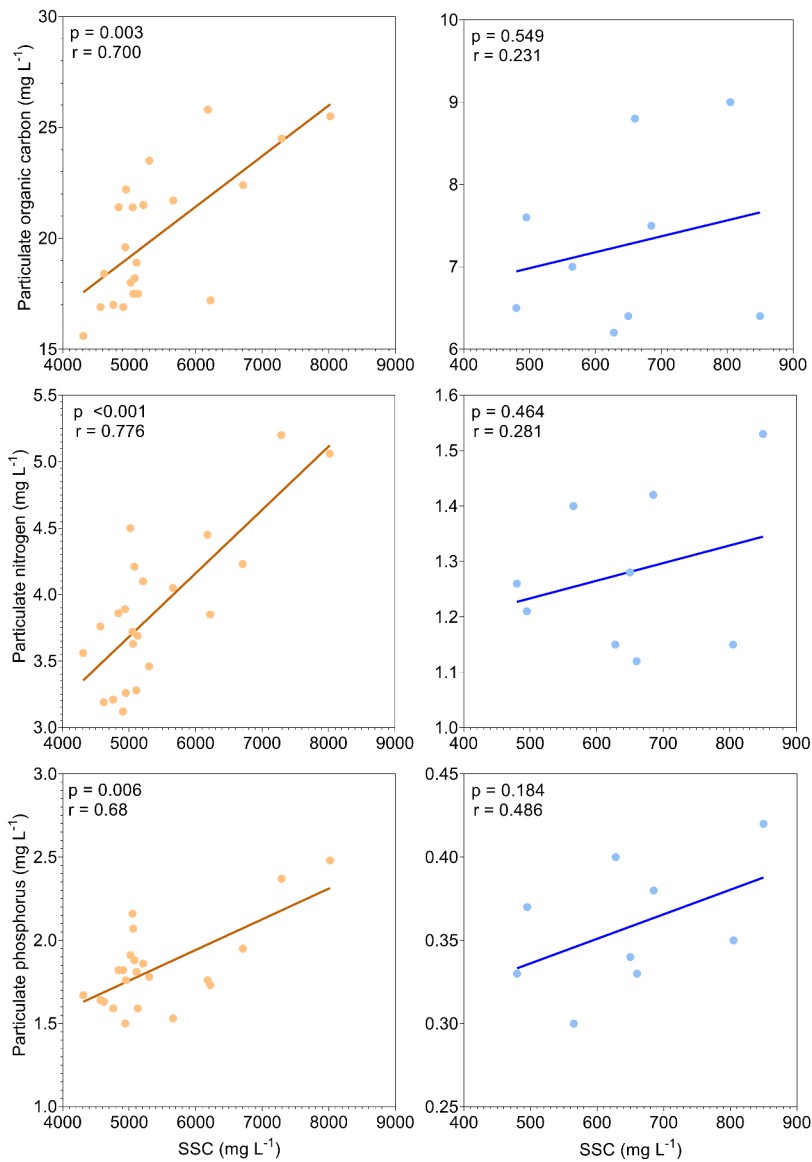

**Figure 10. Relationships between SSC, and POC and nutrient concentrations in the control (brown) and remediated gully (blue) from single flow events on the same day during 2017/2018 wet season.**

### 3.4 Monitoring approach assessment

The large investment in monitoring effort reported in this study was necessary in-order to properly assess the effect of landscape scale remediation on alluvial gully water quality, as well as to test the effectiveness of the different monitoring methods. It is imperative that environmental managers apply robust monitoring plans when conducting gully erosion control measures to ensure their effectiveness is appropriately evaluated. This study identified several important factors to consider when implementing a gully water quality monitoring plan:

(1)      The combination of a small number of high-cost monitoring methods (i.e., autosamplers) complemented by low-cost automated methods (i.e., RS and PASS samplers) allows for both redundancy and more representative


data collection at key monitoring locations, such as gully outlets. For example, the PASS sampler collected samples from events that occurred after the RS and autosamplers were at capacity; the RS samplers provided important information on in-stream suspended sediment heterogeneity over the rising stage; and the autosampler

provided important discrete sample data used to evaluate suspended sediment dynamics (e.g., SSC and water-level hysteresis).

(2) The application of low-cost methods (e.g., the PASS sampler) allows for the establishment of a wider spatial monitoring network. In this study the PASS sampler was deployed at several monitoring locations, in both gully catchments and outlets, which would commonly not be a feasible approach with the other runoff monitoring

methods.

(3) A complete conceptual model of potential inputs and outputs of a gully should be established before monitoring begins. Failure to do so could lead to inconclusive results and a poor evaluation of gully remediation effectiveness. For example, the lack of catchment data for the 2017/2018 wet season needed to be addressed for the following wet seasons in order to account for all the potential influences acting on the suspended sediment

dynamics occurring in the gullies.



## 4 Conclusion

The multiple lines of evidence from this water quality study indicate the application of intensive landscape-scale remediation on actively eroding alluvial gullies has the potential to reduce average suspended sediment
concentrations by more than 80%. This is accompanied by the added benefit of significant reductions in nutrients (nitrogen and phosphorus) and carbon concentrations flowing through the gully. Monitoring was conducted over 2 consecutive wet seasons, however, due to the complications of accurately measuring water velocity in gullies (e.g., rapid channel aggradation or scouring effecting measurement recording) sediment loads could not be accurately estimated within the project timeframe. Further development of gully flow velocity or discharge
measurement capabilities should be conducted to address the current limitations of discharge measurement in these often-remote locations. Furthermore, in-order to understand the effects of landscape scale gully remediation on the reduction of bioavailable nutrient export, future studies should investigate the speciation of particulate and dissolved nutrients in remediated and active alluvial gully systems.

This study has demonstrated the advantages of using multiple suspended sediment monitoring methods in a
configuration that ensures one method is complimentary to the limitations of another. Future water quality studies, particularly in different alluvial gully types, should implement similar monitoring networks to determine if the findings of this study are applicable to the wide range of alluvial gullies in the catchments of the GBR. The intensive remediation of alluvial gullies, as described in this study, represents a promising opportunity to significantly reduce the contribution of sediment and associated nutrients flowing from gullies to the GBR.
However, more information is needed, particularly sediment load estimates and information on remediation longevity over decadal timescales.

## 5 Data and Code availability

The data that support the findings of this study are available from the corresponding author upon reasonable
request.

## 6 Author contribution

- **NJCD:** Designed the experiments, carried them out, sample and data analysis, investigated the data, and prepared the manuscript with contributions from all co-authors.
- **WWB**: Provided guidance in experimental design, method development, interpretation of results, and
provided assistance with preparation of the manuscript.
- **JRS**: Provided assistance in method development, carried out field work, and interpretation of results.
- **AGG and JMB**: Provided sample and data analysis, interpretation of results, and assistance with preparation of the manuscript.
- **PRT and DTW**: Provided interpretation of results, and assistance with preparation of the manuscript.
- **APB**: Provided study supervision, guidance in experimental design, carried out field work, interpretation of results, and assistance with preparation of the manuscript.





## 7 Competing interests

The authors declare that they have no conflict of interest.

## 8 Acknowledgements

The authors acknowledge the traditional owners of the country this study was conducted on. We thank the Laura Ranger group for their support throughout the field monitoring component of this study. We also thank William Higham and Michael Goddard, from Cape York Natural Resource Management, for giving us the opportunity to collaborate with them on the site at Crocodile Station. The authors also acknowledge the Queensland Water

Modelling Network who funded the nutrient analysis component of this study.



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
