# Peer review of "Intensive landscape-scale remediation improves water quality of an alluvial gully located in a Great Barrier Reef catchment"

_Hydrology and Earth System Sciences, 2020_

## Referee Comment (RC1) · Anonymous Referee #1 · 3 Aug 2020

The manuscript hess-2020-268 titled "Landscape scale remediation reduces concentrations of suspended sediment and associated nutrients in alluvial gullies of a Great Barrier Reef catchment: evidence from a novel intensive monitoring approach" has been reviewed. The manuscript is really interesting and fits in the broad scope of the journal. The authors present a detailed comparison analysis of two gullied areas in Australia: One remediated area and one control area. I consider that moderate/major revisions should be carried out before a final decision. Some important questions

should be answered and small issues should be improved.

- One of my main concerns is the limited study period: only two years, and some limitation in sediment samples that were only recorded during three events. The authors stated this problem in the text, but I think that the limitations should be highlighted in the results and discussion section and also in the conclusion section. - I consider that the landscape scale remediation that has been carried out in this area, is really significant to understand all the process, and it should be noted. In that sense, I consider that: - Some photos should be included with situations before and after the reclamation activity (it is included the video, but I consider interesting to include some photos). - In the abstract, you should already inform about the remediate measures. - You should also discuss about the feasibility of this remediation work. Would it be possible to carry out this work in other study areas? Which was the cost of this remediation technique? - I don't really understand the information that you provided in the lines 95-115. Is it about previous remediation activity? I'm not totally sure if this information should be included in this section or it should be moved to the introduction or event results section.

- One really important issue is about literature review and other reclamation examples. In your case, your literature is mainly focused in studies carried out related to the GBR. However, there are other worldwide examples that could be included in the introduction and discussion section to discuss about remediation works. Some examples of remediation can be found in other areas as the Draix catchments (Rey et al. Burylo et al., 2014; Breton et al., 2016) or in Spain (Ballesteros et al. 2017; Oleagordia Montaña et al. 2016). I think that this information could be included and discussed about the feasibility of remediation works in gully and badland areas.

- Other important issue is about the methodology to check the effectiveness of remediation works. You have been mainly focused in this work in turbidity measures, water samples. . . but what about UAVs information? Maybe it could be also interesting to use other kind of instrumentation that can provide different data to complete the dataset. Which are the topographic changes that have been observed in the area?

- I consider that an initial research hypothesis should be included together with the objectives and research questions at the end of the introduction section. - You should specify all the abbreviations and that you have used in the text, in figure and table captions (for example table 1) - In my opinion the tittle is too long.

---

## Referee Comment (RC2) · Simon Walker (Referee) · 10 Aug 2020

Overall, I considered this paper to be a suitable study for HESS and a useful contribution to our knowledge of alluvial gully remediation strategies. In my opinion this is suitable for publication with minor/moderate revisions.

What was the cost of remediation? I think for a global audience this is important.

[Figure]

Placement of devices in the gully catchments: catchment 3 PASS is not on a drainage line but catchment 1 and 2 are (assuming blue lines in Figure 1 are drainage lines found using some routing method?). This seems to have an impact on measured sediment concentrations (for catchment 3 the upper SSC is 3556 while catchment 1 and 2 are 563 and 1517, respectively). Given the focus of the paper is on measurement methods I think a little more discussion about the placement of sensors would be good. I think some more discussion of this is important because it seems to have important implications for your conclusions. Taking the lower end estimate of TWA SSC from the control gully gives 4453 and the upper estimate from the hillslope in catchment 3 is 3556 which is ∼80% of what is seen in the control gully. Without a larger sample it's hard to know whether this is representative or not but for me it suggests the possibility that hillslope erosion, in this environment, is a considerable source of fine sediment (potentially almost equal to gully erosion?). Given that, I think it warrants a little more discussion around possible ways to address the influence of sensor locations with respect to process interpretation.

Also, how do the catchment areas compare? The total catchment area of the remediated gully is ∼13ha but what is the catchment area above each PASS sensor in the sub-catchments and how does this play into the results? And the catchment area for each sub-catchment of the remediated gully.

What other studies have looked at results of similar remediation works on gullies (if any) and what are some of the possible post-remediation issues either currently observable or expected? In other cases (e.g. China, Africa) there is often a focus on gully remediation for land reclamation/conservation purposes more so than sediment runoff reduction. In that case I think there will be more interested in long-term stability of the measures, especially if the cost is high.

If possible, I think a before remediation and after remediation DEM image (or DEM of difference maybe) would be a useful addition.

Figure 7 and 8 seem to suggest that the fine fraction is coming from the catchments more so than the gullies? But there isn't much discussion about this? Maybe I'm interpreting the results wrong but if this is the case, I think it's one of the more interesting findings for discussion.

In your abstract and conclusions you present a value of 80% as the sediment reduction achieved but it's not clear how this number is calculated? Is it the (SSC control – SSC remediated) / (SSC control)? Or some other number?

52: "There are various types of gullies present in the GBR catchment region (e.g., hillslope, colluvial, ephemeral, and soft-rock badlands), however, alluvial gullies likely represent the largest source of sediment, accelerated by land use change, to the GBR." - Reference?

90: "The study site topography is relatively flat." - Would be good to know average slope?

101: "Erosion rates derived from repeated airborne LiDAR between 2009 and 2015 (before remediation activity), indicate the control gully produced slightly more sediment (61 t -1 ha-1 yr-1 ) compared to the remediated gully (50 t-1 ha-1 yr-1 ), based on gully catchment area." - Per unit area of gully or catchment?

102 – 103: t-1 ha-1 yr-1 » t . ha-1 . yr-1 mass shouldn't be a reciprocal here.

103: "Note, LiDAR does not account for the surface erosion generated from the catchment area of each gully, which would be expected to be comparable on an area normalised basis. Hence, the difference in specific yields between the treatment and control would be less than indicated by the LiDAR data alone (Brooks et al., 2016)." - I find this statement a little confusing. I think you either need to be clearer about what this means or not include it.

169: "time weighted average (TWA) SSC" – I can take a guess at what this is but it would be nice to have an equation.

---

## Short Comment (SC2) · 10 Aug 2020

This manuscript addresses a clear knowledge gap: the evaluation of a remediation technique for alluvial gullies in a tropical setting.

The primary conclusion is given as "The multiple lines of evidence from this water quality study indicate the application of intensive landscape-scale remediation on actively eroding alluvial gullies has the potential to reduce average suspended sediment

concentrations by more than 80%."

This conclusion is made on the basis of a comparison of 2 wet seasons of suspended sediment measurement using 4 different types of water samplers for a single control (3.3 hectares catchment area) and single treated catchment (13.7 hectares catchment area) - noting that 3 treatment catchment areas are labelled in figure 1.

The recent review of gully remediation efficacy of Bartley et al. (2020) demonstrates other similar studies have combined multi-year monitoring, pre-treatment measurement and replication in a range of settings. No study was found to provide a "gold standard" BACI, multi-decadal and replicated study but conclusions and attribution were normal reduced as a result.

I looked for "multiple lines of evidence" but only found the suspended sediment sampling with 4 devices arranged side by side.

The two years of sampling does not enable any assessment of whether the hydrological forcing can be interpreted in terms of long term rainfall variability.

Furthermore this sampling does not necessarily represent the long term (decades) performance of the remediation measures.

The difference in performance between the control and treatment gullies is well-summarised in terms of the suspended sediment concentrations and the particle size distributions. The difference between these measures is then attributed to the treatment effect. While this step is intuitive, it is not formally supported given the many limitations of the methodology as noted above.

Reference: Bartley, R., Poesen, J., Wilkinson, S. and Vanmaercke, M., 2020. A review of the magnitude and response times for sediment yield reductions following the rehabilitation of gullied landscapes. Earth Surface Processes and Landforms. Available at https://onlinelibrary.wiley.com/doi/pdf/10.1002/esp.4963

---

## Referee Comment (RC3) · Anonymous Referee #3 · 11 Aug 2020

GENERAL REMARKS The reviewed manuscript refers to the interesting topic on remediation measures used to decrease the negative impact of gully erosion. Such studies are highly needed, especially when they are carried out in one of the most valuable area around the world as the Great Barrier Reef. I appreciate that Authors tested different monitoring methods and evaluated them. These findings may be useful in other areas characterized by dispersive soils and intense short rainfall events.

In my opinion this manuscript fits to the scope of Hydrology and Earth System Sciences journal. The methods are clearly presented (some minor remarks are marked below). The results and conclusions are generally clear, concise, and well-structured. Although, I think that this section can be improved. It would be great to see some comparison of remediation measures used in this study with studies from other regions. The figures are readable and they correspond well with the data presented in supplement.

In order to improve the quality of the paper, I include below some minor remarks.

DETAILED REMARKS

Lines 1-4 Please, consider shortening the title.

Lines 20-21 I suggest to include some information on methods to the abstract. Now you just wrote that novel monitoring network was used without any details.

INTRODUCTION

Can you refer also to the studies on remediation measures in other areas, not only in the GBR catchments?

Line 54 Slacking or slaking?

METHODS

Line 95 I'm confused. You wrote in the text that you used two gullies in the study, whereas in Figure 1 you marked three remediated gully catchments and one control gully catchment. Were these three gully catchments treated as one? Can you mark them together for instance with the same colour line or somehow marked them as one site?

Lines 120-129 I suggest to include some photos from the study area. I know that you present several photos in the supplement, but I think that some of them should be in the manuscript, e.g.., control gully, remediated gully before and after remediation.

Lines 187-192 Did you analyse the whole soil profiles or did you only take samples from the topsoil/subsoils? At which depth did you take samples? Why did you put this subsection (2.4.3. Soil sampling and analysis) into section 2.4. Monitoring methods? I suppose that you did these analyses only once and PSD in soils wasn't monitored.

Line 194 Which samples? I suppose that suspended sediments, but it should be clarified.

---

## Author Comment (AC1) · 29 Sep 2020

GENERAL COMMENTS FROM REFEREE 1:

The manuscript hess-2020-268 titled "Landscape scale remediation reduces concentrations of suspended sediment and associated nutrients in alluvial gullies of a Great Barrier Reef catchment: evidence from a novel intensive monitoring approach" has been reviewed. The manuscript is really interesting and fits in the broad scope of the

journal. The authors present a detailed comparison analysis of two gullied areas in Australia: One remediated area and one control area. I consider that moderate/major revisions should be carried out before a final decision. Some important questions should be answered and small issues should be improved.

RESPONSE: Acknowledge.

The authors acknowledge this positive comment and have undertaken specific reconsideration to address the other key points raised by the Referee(see below).

SPECIFIC COMMENTS FROM REVIEWER 1:

Specific Comment 1. "One of my main concerns is the limited study period: only two years, and some limitation in sediment samples that were only recorded during three events. The authors stated this problem in the text, but I think that the limitations should be highlighted in the results and discussion section and also in the conclusion section. - I consider that the landscape scale remediation that has been carried out in this area, is really significant to understand all the process, and it should be noted. In that sense, I consider that: - Some photos should be included with situations before and after the reclamation activity (it is included the video, but I consider interesting to include some photos). - In the abstract, you should already inform about the remediate measures. - You should also discuss about the feasibility of this remediation work. Would it be possible to carry out this work in other study areas? Which was the cost of this remediation technique? – I don't really understand the information that you provided in the lines 95-115. Is it about previous remediation activity? I'm not totally sure if this information should be included in this section or it should be moved to the introduction or event results section.

RESPONSE: Accept/Clarify.

The authors accept the suggestions by the referee and wish to clarify the following points:
Two-year study period:

The authors agree with the referee regarding the relatively short timeframe of monitoring. However, the project was only funded for this time period. The authors mention in the abstract that the study represents a preliminary evaluation (Line 18) for this reason. The authors agree that more emphasis should be placed on this factor and will include discussion of this limitation in the Results, Discussion, and Conclusion sections.

Sediment samples collection:

The Referee indicates that sediment samples "were only recorded during three [flow] events". A total of 207 suspended sediment samples were collected from 19 flow events during the study period. Therefore, the authors interpret that the referee's comment is specifically referring to the collection of samples for nutrient analysis, of which three flow events were monitored. The authors agree with the Referee that three flow events is a relatively low number of flow events and will provide more commentary regarding this limitation and further explain the logistical and safety challenges that makes collecting a large number of nutrient samples from the site infeasible. However, the authors believe that the sample numbers collected from the three flow events (Remediated gully n=14 and Control Gully n =26) are sufficient for conducting a preliminary evaluation of nutrient water quality trends. The authors will revise the text to include the number of samples collected for nutrient analysis. Before and after photos: The authors agree that before and after photos of the Remediated and Control gullies are important. These photos are referenced throughout the document and are included in the supplementary information. The authors accept that some of these before and after photos should be included as figures in the manuscript.

More detail on remediation in Abstract:

The authors agree with this comment and will provide more specific information regarding the erosion controls used as part of the remediation process in the abstract of the manuscript.

In the abstract mention of feasibility, cost of remediation, and suitability for this remediation to be used in other study areas should be included:

The Referee makes some good points regarding the logistical considerations of landscape scale gully remediation. The authors accept that these features should be briefly described throughout the manuscript to provide context. However, these features should not be treated as key components for meeting the objective of the study, which is to evaluate the effectiveness of remediation on gully water quality using novel monitoring techniques.

Mention cost of remediation in Abstract:

The authors agree that description of the cost of remediation will provide context regarding the importance of the remediation activities completed and the need to monitor their effectiveness.

Unclear description of gullies in Methods (Lines 95-115): The Referee comments on a section of text in the methods that describes the physical/geomorphological features of the gullies used as part of the study. The authors accept that this section of the paragraph can be improved, however, we believe that its location in the Study site section of the Methods section is appropriate. The authors will revise the text to provide a clearer description of features and estimated erosion rates of the gullies used for the study.

Specific Comment 2. "One really important issue is about literature review and other reclamation examples. In your case, your literature is mainly focused in studies carried out related to the GBR. However, there are other worldwide examples that could be included in the introduction and discussion section to discuss about remediation works. Some examples of remediation can be found in other areas as the Draix catchments (Rey et al. Burylo et al., 2014; Breton et al., 2016) or in Spain (Ballesteros et al. 2017; Oleagordia Montaña et al. 2016). I think that this information could be included and discussed about the feasibility of remediation works in gully and badland areas."

RESPONSE: Acknowledge.

The authors acknowledge that the literature review may focus too much on studies completed in catchments of the Great Barrier Reef and will include discussion on relevant remediation works completed elsewhere to provide a more global context to the study. The authors thank the referee for the suggested examples and will consider them when revising the literature review.

Specific Comment 3. "Other important issue is about the methodology to check the effectiveness of remediation works. You have been mainly focused in this work in turbidity measures, water samples: but what about UAVs information? Maybe it could be also interesting to use other kind of instrumentation that can provide different data to complete the dataset. Which are the topographic changes that have been observed in the area?"

RESPONSE: Acknowledge.

The authors acknowledge that remote sensing methods used to estimate erosion (i.e., airborne Lidar) could provide a useful complimentary line of evidence for the study. The authors provide discussion points and relevant data derived from terrestrial and airborne Lidar measurements throughout the manuscript. However, the authors did not extensively compare the results of the two methods in more detail as this would make the manuscript less concise and is not related to the overall aim of the study (i.e. water quality aspects of gully remediation).

Specific Comment 4. "I consider that an initial research hypothesis should be included together with the objectives and research questions at the end of the introduction section. - You should specify all the abbreviations and that you have used in the text, in figure and table captions (for example table 1) - In my opinion the tittle is too long."

RESPONSE: Acknowledge.

The aim of the study is described in the final paragraph of the Introduction (Lines

76-78). The authors will revise this text to make the research questions and objectives clearer. The authors understand that there are many acronyms and abbreviations throughout the manuscript. However, we believe they are all specified when first mentioned in the text. The authors will consult with the Journal editor if an index of abbreviations and acronyms is appropriate for the manuscript. The authors will consider reducing the length of the title to make it more concise.
* * *

---

## Author Comment (AC2) · 29 Sep 2020

GENERAL COMMENTS FROM PETER HAIRSINE:

General Comments 1. "This manuscript addresses a clear knowledge gap: the evaluation of a remediation technique for alluvial gullies in a tropical setting."

RESPONSE: Acknowledge.

[Figure]

The authors acknowledge this positive comment and have undertaken specific reconsideration to address the other key points raised by Peter Hairsine (see below).

SPECIFIC COMMENTS FROM PETER HAIRSINE:

Specific Comment 1. "The primary conclusion is given as "The multiple lines of evidence from this water quality study indicate the application of intensive landscape-scale remediation on actively eroding alluvial gullies has the potential to reduce average suspended sediment concentrations by more than 80%." This conclusion is made on the basis of a comparison of 2 wet seasons of suspended sediment measurement using 4 different types of water samplers for a single control (3.3 hectares catchment area) and single treated catchment (13.7 hectares catchment area) - noting that 3 treatment catchment areas are labelled in figure 1." The recent review of gully remediation efficacy of Bartley et al. (2020) demonstrates other similar studies have combined multi-year monitoring, pre-treatment measurement and replication in a range of settings. No study was found to provide a "gold standard" BACI, multi-decadal and replicated study but conclusions and attribution were normal reduced as a result."

RESPONSE: Clarify.

The authors thank Peter for notifying the authors of the recent study completed by his colleagues (Bartely et al., 2020). This manuscript was submitted to HESS prior to the publication of the Review article Peter mentions. The authors will include the findings of the review in the revised manuscript literature review.

Specific Comment 2. "I looked for "multiple lines of evidence" but only found the suspended sediment sampling with 4 devices arranged side by side."

RESPONSE: Acknowledge/Clarify.

For context, Peter is referring to Line 460 "The multiple lines of evidence from this water quality study indicate the application of intensive landscape-scale remediation on actively eroding alluvial gullies has the potential to reduce average suspended sediment
concentrations by more than 80%." In this statement the authors are referring to the collection of water quality data using different monitoring methods (i.e., four different methods for collecting water quality data) to assess the effect of gully remediation on water quality. Each of the water quality monitoring methods used collect a sample in a manner that is independent compared to the others, thus, evidence is provided by four separate lines of data gathering. Furthermore, the use of different water quality analyses provides further relevant lines of evidence, that are complimentary to the separate collection methods, (i.e., suspended sediment concentration, particle size distribution, and nutrient and carbon analyses). The authors acknowledge that the term "multiple lines of evidence" may mislead some readers to thinking that complimentary data (i.e., Lidar soil loss estimates) are referenced here. Thus, the authors will revise the text to state "The water quality data collected during this study, using multiple monitoring methods, indicate the application of intensive landscape-scale remediation on actively eroding alluvial gullies has the potential to reduce average suspended sediment concentrations by more than 80%."

Specific Comment 3. "The two years of sampling does not enable any assessment of whether the hydrological forcing can be interpreted in terms of long term rainfall variability. Furthermore this sampling does not necessarily represent the long term (decades) performance of the remediation measures."

RESPONSE: Acknowledge.

The authors agree with Peter in that there is a need for more monitoring data, over longer time scales, to evaluate the effects of long term stressors (i.e., rainfall variability and backwater flooding effects). The authors infer this sentiment in the final statement of the conclusion section Line 477: " However, more information is needed, particularly sediment load estimates and information on remediation longevity over decadal timescales." The authors believe this important point should also be mentioned in the Abstract and will include a statement that emphasises the relatively short timescale that the study was conducted over (i.e., two years) and how more monitoring data is

needed to assess the long term durability of the remediation measures used.

Specific Comment 4. "The difference in performance between the control and treatment gullies is well summarised in terms of the suspended sediment concentrations and the particle size distributions. The difference between these measures is then attributed to the treatment effect. While this step is intuitive, it is not formally supported given the many limitations of the methodology as noted above."

RESPONSE: Clarify.

The aim of the study was to determine if gully water quality conditions were improved by landscape-scale remediation using water quality monitoring methods that employ sampling processes that are unique from one another. This was done so that limitations of one or more of the monitoring methods used could be accounted for when evaluating the effectiveness of the remediation measures. The water quality of the overland flow waters draining into the Remediated and Control gullies from their respective catchments is relatively similar. The overland flow water represents the major transport mechanism for suspended sediment within the gully system. Thus, the lower concentrations of suspended sediment and associated nutrients and carbon in the Remediated gully compared to the Control gully can only be attributed to the reduction in sediment and nutrient sources (i.e., erodible soil) from within the gully itself. This is not an intuitive assessment, rather, it is an interpretation of the data gathered. Furthermore, the authors shall provide a before and after digital elevation map (DEM) of the remediated gully that demonstrates how erosion of the gully system has been greatly reduced. This complimentary line of evidence will further support the conclusions made regarding the effectiveness of the remediation measures used at the Remediated gully.

---

## Author Comment (AC3) · 29 Sep 2020

Please refer to the authors response for SC1.

---

## Author Comment (AC4) · 29 Sep 2020

GENERAL COMMENTS FROM REFEREE 2:

General Comments 1. Overall, I considered this paper to be a suitable study for HESS and a useful contribution to our knowledge of alluvial gully remediation strategies. In my opinion this is suitable for publication with minor/moderate revisions.

[Figure]

RESPONSE: Acknowledge.

The authors acknowledge this positive comment and have undertaken specific reconsideration to address the other key points raised by the Referee (see below).

SPECIFIC COMMENTS FROM REFEREE 2:

Specific Comment 1. "What was the cost of remediation? I think for a global audience this is important."

RESPONSE: Accept.

Note, Referee 1 also mentioned this as an important factor to include in the manuscript. The authors agree that the cost of remediation provides important context for the global audience of the Journal and will include discussion regarding the cost of remediation in the revised manuscript.

Specific Comment 2. "Placement of devices in the gully catchments: catchment 3 PASS is not on a drainage line but catchment 1 and 2 are (assuming blue lines in Figure 1 are drainage lines found using some routing method?). This seems to have an impact on measured sediment concentrations (for catchment 3 the upper SSC is 3556 while catchment 1 and 2 are 563 and 1517, respectively). Given the focus of the paper is on measurement methods I think a little more discussion about the placement of sensors would be good. I think some more discussion of this is important because it seems to have important implications for your conclusions. Taking the lower end estimate of TWA SSC from the control gully gives 4453 and the upper estimate from the hillslope in catchment 3 is 3556 which is ~80% of what is seen in the control gully. Without a larger sample it's hard to know whether this is representative or not but for me it suggests the possibility that hillslope erosion, in this environment, is a considerable source of fine sediment (potentially almost equal to gully erosion?). Given that, I think it warrants a little more discussion around possible ways to address the influence of sensor locations with respect to process interpretation."

RESPONSE: Acknowledge/clarify.

The Referee makes a good point regarding the need for more detailed description of overland flow sampling methodology. Please note that the blue draining lines show in Figure 1 are only indicative of the actual overland flow characteristics observed at the site. Other factors (e.g., vegetation, natural debris, and termite mounds) influence water flows in ways that are not perceived by airborne Lidar-derived flow lines. The overland flow sampling locations were chosen based on observing locations that had consistent flows and were as close as possible to the transition of catchment to gully. The sampler located at catchment three was placed in a different location to the drainage line indicated in Figure 1 because of the presence of termite mounds and vegetation. The authors thank the Referee for making this observation and will provide commentary in the caption of Figure 1 to provide context for the blue flow drainage lines. The authors will also revise the text in the methods (Lines 181-185) to include more detail and photographs of overland flow at the sampling locations (these will be provided as supplementary information). The referee makes a good point by comparing the suspended sediment dynamics of the catchments and gullies, specifically for catchment vs. gully sediment sources in the control gully. The authors caution the Referee against comparing catchment/gully suspended sediment sample concentrations collected from different locations (i.e., comparing the remediated gully catchment to the control gully outlet). This is not appropriate given there is suspended sediment sample data collected from a location that represents the majority of the catchment water draining into the Control gully. Furthermore, Section 3.2.2 Relationship between SSC and flow provides discussion and detailed examples from the data indicating that subsurface erosion processes are the dominant source of suspended sediment flowing from the Control gully. However, the Referee makes a good point that erosion processes in the catchment, possibly sourced from surface erosion, appear to be a major contributor of suspended sediment flowing through the gully systems and that the collection of accurate and representative catchment monitoring data is very important to understanding these dynamics. The authors will revise the existing commentary on

catchment suspended sediment contribution, in the Results and Discussion and Conclusions sections, to provide emphasis on the importance of monitoring locations for the purpose of collecting representative catchment overland flow samples.

Specific comment 3. " Also, how do the catchment areas compare? The total catchment area of the remediated gully is ~13ha but what is the catchment area above each PASS sensor in the sub-catchments and how does this play into the results? And the catchment area for each sub-catchment of the remediated gully."

RESPONSE: Acknowledge.

The aim of measuring suspended sediment in water flowing overland into the gullies was to understand their contribution, in terms of suspended sediment concentration and particle size distribution, to the suspended sediment measured at the gully outlet. Because of this, the size of the catchment is less important as ensuring that the major catchment drainage inputs into the gully are monitored. For example, the remediated gully catchment drains into the gully from three separate locations, later mixing at a confluence within the gully. Thus, three monitoring locations were required to account for the majority of overland flows draining from the catchment. In contrast, the majority of catchment overland flows into the control gully drain through one location, thus, it was monitored at one location up-stream of the gully head. Evaluation of the influence of sub-catchment area on the contribution of suspended sediment to the gullies would require the estimation of suspended sediment loads from these sub-catchments. As discussed in the manuscript (Lines 290-295 and 463-466) the estimation of loads from these highly ephemeral systems, and their catchments, is very challenging and was not feasible for this study. The authors acknowledge the importance of the catchment area regarding overland flow sediment contributions and will provide commentary on this in the Results and Discussion section.

Specific comment 4 . "If possible, I think a before remediation and after remediation DEM image (or DEM of difference maybe) would be a useful addition."

RESPONSE: Accept.

Note, Referee 1 mentioned that before and after photos of the remediated gully would be beneficial to the manuscript. The authors agree that a before and after digital elevation map (DEM) would also be beneficial. The authors will include a figure with before and after photos and DEM images of the remediated gully in the revised manuscript.

Specific comment 5 . "Figure 7 and 8 seem to suggest that the fine fraction is coming from the catchments more so than the gullies? But there isn't much discussion about this? Maybe I'm interpreting the results wrong but if this is the case, I think it's one of the more interesting findings for discussion."

RESPONSE: Clarify.

Figure 7 and 8 demonstrate that the distribution of fine sediment (0.1-30 $\mu$m) in the suspended sediment samples collected from the catchments and gully outlets are similar. However, further investigation (e.g., geochemical tracing) would be required to differentiate the sources of fine sediment in the gully outlet sample. The authors agree with the referee's observation that the catchments samples appear to consist of mostly fine sediment as is indicated by in Figures 7 and 8 and Table 3 (d90 < 36 $\mu$m for all overland flow samples collected). Given this it could be suggested that the catchments contribute some of the fine suspended sediment measured at the gully outlet. This is discussed in Section 3.2.3 Particle size distribution, Lines 356-368: "The suspended sediment PSD characteristics of control gully catchment PASS samples was notably different to the gully outlet PASS samples (Table 3). This indicates the contribution of slightly coarser suspended sediment from gully erosion (d50 10.8 $\mu$m) is greater than the suspended sediment contribution of the catchment (d50 4.29 $\mu$m) in the control gully. In contrast, the PSD characteristics of suspended sediment samples collected from the outlet of the remediated gully (d50 of 5.84 $\mu$m) and samples collected from Catchments 2 (d50 of 5.52 $\mu$m) and 3 (d50 of 5.06 $\mu$m) of the three catchment areas draining into the gully were very similar (Table 3) (Figure 8). This suggests there is a

notable contribution of sediment entering both gullies from their respective catchments."

Specific comment 6. "In your abstract and conclusions you present a value of 80% as the sediment reduction achieved but it's not clear how this number is calculated? Is it the (SSC control – SSC remediated) / (SSC control)? Or some other number?"

RESPONSE: Clarify.

The sections the Referee mentions, state the following: Lines 22-23: "Suspended sediment concentrations were ~80% lower at the remediated site compared to the control site,..." and Lines 460-463: "The multiple lines of evidence from this water quality study indicate the application of intensive landscape-scale remediation on actively eroding alluvial gullies has the potential to reduce average suspended sediment concentrations by more than 80%." These statements imply that the SSCs of the different gullies were compared and the difference in concentration between the two was ~80%. This comparison is discussed in further detail in Section 3.2.1 Suspended sediment concentration. It is not uncommon to see statements such as these without detailed explanations of the exact formula used in the abstract conclusion sections of a scientific journal article.

Specific comment 7. "[Line] 52: "There are various types of gullies present in the GBR catchment region (e.g., hillslope, colluvial, ephemeral, and soft-rock badlands), however, alluvial gullies likely represent the largest source of sediment, accelerated by land use change, to the GBR." - Reference?"

RESPONSE: Clarify.

The text immediately following this sentence uses the same reference. Thus, the reference is provided at the end of the paragraph. The references used are: Brooks et al., 2013; Brooks et al., 2016; Brooks et al., 2019. The authors acknowledge that some time has passed since this the submission of this manuscript and shall review the scientific literature for findings that support or contradict the statement.

Specific comment 8. " [Line] "90: "The study site topography is relatively flat." - Would be good to know average slope?"

RESPONSE: Accept.

The authors agree that description of the average slope of the site would help with visualisation of the site conditions and will ensure it is included in the revised manuscript.

Specific comment 9. " 101: "Erosion rates derived from repeated airborne LiDAR between 2009 and 2015 (before remediation activity), indicate the control gully produced slightly more sediment (61 t -1 ha-1 yr-1 ) compared to the remediated gully (50 t-1 ha-1 yr-1 ), based on gully catchment area." - Per unit area of gully or catchment?"

RESPONSE: Clarify. The authors reference the sediment yields estimated by Brooks et al., 2016, where the unit area was inclusive of the gully and its associated catchment area.

Specific comment 10. [Line] "102 – 103: t-1 ha-1 yr-1 Âż t . ha-1 . yr-1 mass shouldn't be a reciprocal here.

RESPONSE: Accept.

The authors accept this comment and will revise the qualifiers in the text to read x number of t/ha/yr (i.e., x number of tonnes per hectare per year).

Specific comment 11. [Line] "103: "Note, LiDAR does not account for the surface erosion generated from the catchment area of each gully, which would be expected to be comparable on an area normalised basis. Hence, the difference in specific yields between the treatment and control would be less than indicated by the LiDAR data alone (Brooks et al., 2016)." - I find this statement a little confusing. I think you either need to be clearer about what this means or not include it.

RESPONSE: Accept.

The authors included this statement to provide context, in the form of empirical data,

that the gullies, normalised for area, were contributing comparable sediment yields. The authors thank the Referee for the observation and will ensure this statement is clearer in the revised manuscript.

Specific comment 12. [Line 169]: "time weighted average (TWA) SSC" – I can take a guess at what this is but it would be nice to have an equation.

RESPONSE: Accept.

The authors thank the Referee for pointing out this oversight and will ensure an example formula for the calculation of time-weighted suspended sediment concentration is included in the Methods section.

---

## Author Comment (AC5) · 29 Sep 2020

GENERAL COMMENTS FROM REFEREE 3:

General Comments 1. "GENERAL REMARKS The reviewed manuscript refers to the interesting topic on remediation measures used to decrease the negative impact of gully erosion. Such studies are highly needed, especially when they are carried out in one of the most valuable area around the world as the Great Barrier Reef. I appreciate

that Authors tested different monitoring methods and evaluated them. These findings may be useful in other areas characterized by dispersive soils and intense short rainfall events. In my opinion this manuscript fits to the scope of Hydrology and Earth System Sciences journal. The methods are clearly presented (some minor remarks are marked below). The results and conclusions are generally clear, concise, and well-structured. Although, I think that this section can be improved. It would be great to see some comparison of remediation measures used in this study with studies from other regions. The figures are readable, and they correspond well with the data presented in supplement. In order to improve the quality of the paper, I include below some minor remarks."

RESPONSE: Acknowledge.

The authors acknowledge this positive comment and have undertaken specific reconsideration to address the other key points raised by the Referee (see below).

SPECIFIC COMMENTS FROM REFEREE 3:

Specific Comment 1. " Lines 1-4 Please, consider shortening the title."

RESPONSE: Accept.

Note, Referee 1 also commented on shortening the title. The authors will revise the title to be more concise.

Specific Comment 2. "Lines 20-21 I suggest to include some information on methods to the abstract. Now you just wrote that novel monitoring network was used without any details."

RESPONSE: Accept.

The authors agree with the Referee's comment and will include a more detailed description of the water quality monitoring network in the Abstract.

Specific Comment 3. "Can you refer also to the studies on remediation measures in

other areas, not only in the GBR catchments?"

RESPONSE: Accept.

Referee 1 also comments on the need for more gully remediation examples, from areas outside of the Great Barrier Reef Catchment, in the Introduction. The authors agree that addition of such examples will give the manuscript more of a global context and will include them in the revised version.

Specific Comment 4. "Line 54 Slacking or slaking?"

RESPONSE: Clarify.

The authors thank the Referee for pointing out this oversight. We believe the correct term is slaking sediments. This will be corrected in the revised manuscript.

Specific Comment 5. "Line 95 I'm confused. You wrote in the text that you used two gullies in the study, whereas in Figure 1 you marked three remediated gully catchments and one control gully catchment. Were these three gully catchments treated as one? Can you mark them together for instance with the same colour line or somehow marked them as one site?"

RESPONSE: Clarify/Accept.

The study focused on two gullies and their respective catchments. The catchment of the remediated gully is characterised by three separate sub-catchments that flow into the gully at three distinct locations. In contrast, the majority of catchment drainage into the Control gully occurs at one location and thus, represents one catchment. The authors will revise the catchment boundaries in Figure one to reflect this statement. Commentary regarding this will also be noted in the Figure caption.

Specific Comment 6. "Lines 120-129 I suggest to include some photos from the study area. I know that you present several photos in the supplement, but I think that some of them should be in the manuscript, e.g.., control gully, remediated gully before and

after remediation."

RESPONSE: Accept.

Referee's 1 and 2 also commented on the need for before and after photos of the remediated gully to be included as a figure in the manuscript. The authors agree with the Referee's comments and will include before and after photos in the revised manuscript.

Specific Comment 7. "Lines 187-192 Did you analyse the whole soil profiles or did you only take samples from the topsoil/subsoils? At which depth did you take samples? Why did you put this subsection (2.4.3. Soil sampling and analysis) into section 2.4. Monitoring methods? I suppose that you did these analyses only once and PSD in soils wasn't monitored."

RESPONSE: Acknowledge/Clarify.

Whole soil samples were analysed for particle size distribution using hydrometer techniques. Soil samples were collected from the face the gully (i.e., the areas undergoing erosion) at depths ranging from the surface to 1 m. The soil sampling and analysis section was written as a separate section because these analyses were only conducted once and the authors thought it best not to group it under the water quality monitoring methods section. The authors will include a more detailed description of the soil sampling methods in the revised manuscript.

Specific Comment 8. "Line 194 Which samples? I suppose that suspended sediments, but it should be clarified."

RESPONSE: Accept/clarify.

The authors thank the Referee for pointing out this oversight. The sentence will state the following in the revised manuscript "Water samples collected from the Remediated and Control gullies were analysed for suspended sediment concentration (ASTM standard method D 3977-97), and particle size distribution using laser diffraction spectroscopy (Malvern Mastersizer 3000, Malvern Instruments)."

---

## Author Response (AR1)

**GENERAL COMMENTS FROM REFEREE 1:**

The manuscript hess-2020-268 titled "Landscape scale remediation reduces concentrations of suspended sediment and associated nutrients in alluvial gullies of a Great Barrier Reef catchment: evidence from a novel intensive monitoring approach" has been reviewed. The manuscript is really interesting and fits in the broad scope of the journal. The authors present a detailed comparison analysis of two gullied areas in Australia: One remediated area and one control area. I consider that moderate/major revisions should be carried out before a final decision. Some important questions should be answered and small issues should be improved.

> *RESPONSE:* ***Acknowledge****.* The authors acknowledge this positive comment and have undertaken specific reconsideration to address the other key points raised by the reviewer (see below).

**SPECIFIC COMMENTS FROM REVIEWER 1:**

**Specific Comment 1. "**One of my main concerns is the limited study period: only two years, and some limitation in sediment samples that were only recorded during three events. The authors stated this problem in the text, but I think that the limitations should be highlighted in the results and discussion section and also in the conclusion section. - I consider that the landscape scale remediation that has been carried out in this area, is really significant to understand all the process, and it should be noted. In that sense, I consider that: - Some photos should be included with situations before and after the reclamation activity (it is included the video, but I consider interesting to include some photos). - In the abstract, you should already inform about the remediate measures. - You should also discuss about the feasibility of this remediation work. Would it be possible to carry out this work in other study areas? Which was the cost of this remediation technique? – I don't really understand the information that you provided in the lines 95-115. Is it about previous remediation activity? I'm not totally sure if this information should be included in this section or it should be moved to the introduction or event results section.

*RESPONSE:* ***Accept/Clarify.*** The authors accept the suggestions by the referee and wish to clarify the following points:

**Two-year study period:** Accept. New text has been added to the Results and Discussion and Conclusion sections regarding this:

***Line 319-321***: "However, because this study only includes two wet seasons of data it should be considered preliminary until it is further validated by continued monitoring of the remediated gully for several additional wet seasons."

***Line 506-509***: "Further monitoring at the site should be conducted over longer timescales (i.e., decades) to evaluate the longevity of the erosion mitigation controls used as part of the gully remediation works."

**Sediment samples collection:** Accept. The text has been revised:

***Line 434-438***: "A total of 40 samples were collected from the remediated (n=14) and control (n=26) gullies for nutrient analysis. The hydrographs and SSC trends of these sampling events indicate they were representative of the other flow events observed in the two gullies (SI-3) and provide enough data for a preliminary assessment of nutrient transport trends within the gullies."

**Before and after photos:** Please refer to Figure 2

**More detail on remediation in Abstract:** Accept:

***Line 20-24***: "The gully remediation method was a first attempt, in the region, to invest a high level of financial (total cost of remediation ~$90,000 AUD) and logistical effort (e.g., intensive earthworks including the establishment of an on-site quarry) to develop long lasting erosion mitigation measures (i.e., regraded, compacted and battered gully walls, rock armouring of banks and channel, and installation of rock check dams)."

**Mention cost of remediation in Abstract:** Accept. Please refer to the comment above.

**Unclear description of gullies in Methods (Lines 95-115):** Accept. The text now states:

***Line 118-132***: "Two gullies were used to evaluate the effectiveness of the remediation works. The remediated gully is the larger of the two which encompasses several gully lobes that drain into a central channel. The gully treatment area is ~0.6 ha, with a catchment area of 13.7 ha. The catchment of the remediated gully is a conglomerate of three sub-catchments (Sub-catchments 1-3) (Figure 1). The active secondary incision of the control gully is ~0.2 ha while the gully catchment, which drains directly into the headscarp, has an area of 3.3 ha. Both gullies are situated in highly-dispersible and slaking sodic alluvium. Prior to remediation, both gully catchments would have undergone similar erosion processes (i.e., scalding, sheet erosion, rilling in the gully catchment and tunnel erosion, head scarp mass-failure, and gully sidewall erosion within the incised part of the gully). Erosion rates derived from repeated airborne LiDAR collected before remediation was conducted (2009 to 2015), indicate the control gully produced slightly more sediment (61 t/ha/yr) compared to the remediated gully (50 t/ha/yr), based on gully catchment area (Brooks et al., 2016). Comparison of particle size distribution (PSD) (Section 3.2.3) from readily erodible soil collected from the two gullies, prior to remediation activities, showed there was no significant difference between the two gullies. It is likely that these soils would have eroded into suspended sediment in a similar manner, thus, it is assumed that the suspended sediment concentration (SSC) and PSD from the two gullies would have been similar, pending any significant differences in water velocity."

**Specific Comment 2.** "One really important issue is about literature review and other reclamation examples. In your case, your literature is mainly focused in studies carried out related to the GBR. However, there are other worldwide examples that could be included in the introduction and discussion section to discuss about remediation works. Some examples of remediation can be found in other areas as the Draix catchments (Rey et al. Burylo et al., 2014; Breton et al., 2016) or in Spain (Ballesteros et al. 2017; Oleagordia Montaña et al. 2016). I think that this information could be included and discussed about the feasibility of remediation works in gully and badland areas."

> *RESPONSE:* **_Acknowledge_**. The authors acknowledge that the literature review may focus too much on studies completed in catchments of the Great Barrier Reef and will include discussion on relevant remediation works completed elsewhere to provide a more global context to the study. The authors thank the referee for the suggested examples and will consider them when revising the literature review.

**Specific Comment 3.** "Other important issue is about the methodology to check the effectiveness of remediation works. You have been mainly focused in this work in turbidity measures, water samples: but what about UAVs information? Maybe it could be also interesting to use other kind of instrumentation that can provide different data to complete the dataset. Which are the topographic changes that have been observed in the area?"

> *RESPONSE:* **_Acknowledge_**. The authors acknowledge that remote sensing methods used to estimate erosion (i.e., airborne Lidar) could provide a useful complimentary line of evidence for the study. The authors provide discussion points and relevant data derived from terrestrial and airborne Lidar measurements throughout the manuscript. However, the authors did not extensively compare the results of the two methods in more detail as this would make the manuscript less concise and is not related to the overall aim of the study (i.e. water quality aspects of gully remediation).

**Specific Comment 4.** "I consider that an initial research hypothesis should be included together with the objectives and research questions at the end of the introduction section. - You should specify all the abbreviations and that you have used in the text, in figure and table captions (for example table 1) - In my opinion the tittle is too long."

> *RESPONSE:* **_Acknowledge_**. The aim of the study is described in the final paragraph of the Introduction (Lines 76-78). The authors will revise this text to make the research questions and objectives clearer.
>
> The objectives now state:
>
> ***Line 99-104:*** "Here we aim to assess the effectiveness of landscape-scale remediation in improving the water quality of an alluvial gully situated in the tropics of Queensland, Australia, which flows to the Great Barrier Reef. We apply a recently developed gully water quality monitoring approach that facilitates accurate measurements while meeting the financial and operational requirements of monitoring in remote locations. This work, although done on a limited spatial and temporal scale, provides a critical foundation for developing and evaluating landscape-scale remediation of alluvial gullies in the Great Barrier Reef region."
>
> The title of the manuscript has been revised, it now states: "Intensive landscape-scale remediation improves water quality of an alluvial gully located in a Great Barrier Reef catchment".

**COMMENTS FROM REFEREE 2:**

**GENERAL COMMENTS FROM REFEREE 2:**

**General Comments 1.** Overall, I considered this paper to be a suitable study for HESS and a useful contribution to our knowledge of alluvial gully remediation strategies. In my opinion this is suitable for publication with minor/moderate revisions.

> *RESPONSE:* ***Acknowledge****.* The authors acknowledge this positive comment and have undertaken specific reconsideration to address the other key points raised by the reviewer (see below).

**SPECIFIC COMMENTS FROM REFEREE 2:**

**Specific Comment 1.** "What was the cost of remediation? I think for a global audience this is important.”

> *RESPONSE:* ***Accept****.* Text has been revised in abstract:
>
> ***Line 20-24***: "The gully remediation method was a first attempt, in the region, to invest a high level of financial (total cost of remediation ~$90,000 AUD) and logistical effort (e.g., intensive earthworks including the establishment of an on-site quarry) to develop long lasting erosion mitigation measures (i.e., regraded, compacted and battered gully walls, rock armouring of banks and channel, and installation of rock check dams)."

**Specific Comment 2.** "Placement of devices in the gully catchments: catchment 3 PASS is not on a drainage line but catchment 1 and 2 are (assuming blue lines in Figure 1 are drainage lines found using some routing method?). This seems to have an impact on measured sediment concentrations (for catchment 3 the upper SSC is 3556 while catchment 1 and 2 are 563 and 1517, respectively). Given the focus of the paper is on measurement methods I think a little more discussion about the placement of sensors would be good. I think some more discussion of this is important because it seems to have important implications for your conclusions. Taking the lower end estimate of TWA SSC from the control gully gives 4453 and the upper estimate from the hillslope in catchment 3 is 3556 which is ~80% of what is seen in the control gully. Without a larger sample it's hard to know whether this is representative or not but for me it suggests the possibility that hillslope erosion, in this environment, is a considerable source of fine sediment (potentially almost equal to gully erosion?). Given that, I think it warrants a little more discussion around possible ways to address the influence of sensor locations with respect to process interpretation."
* * *
*RESPONSE:* ***Acknowledge/clarify.*** The Referee makes a good point regarding the need for more detailed description of overland flow sampling methodology. Please note that the blue draining lines show in Figure 1 are only indicative of the actual overland flow characteristics observed at the site. Other factors (e.g., vegetation, natural debris, and termite mounds) influence water flows in ways that are not perceived by airborne Lidar-derived flow lines. The overland flow sampling locations were chosen based on observing locations that had consistent flows and were as close as possible to the transition of catchment to gully. The sampler located at catchment three was placed in a different location to the drainage line indicated in Figure 1 because of the presence of termite mounds and vegetation. The authors thank the Referee for making this observation and will provide commentary in the caption of Figure 1 to provide context for the blue flow drainage lines. The authors will also revise the text in the methods (Lines 181-185) to include more detail and photographs of overland flow at the sampling locations (these will be provided as supplementary information).

The referee makes a good point by comparing the suspended sediment dynamics of the catchments and gullies, specifically for catchment vs. gully sediment sources in the control gully. The authors caution the Referee against comparing catchment/gully suspended sediment sample concentrations collected from different locations (i.e., comparing the remediated gully catchment to the control gully outlet). This is not appropriate given there is suspended sediment sample data collected from a location that represents the majority of the catchment water draining into the Control gully. Furthermore, Section 3.2.2 Relationship between SSC and flow provides discussion and detailed examples from the data indicating that subsurface erosion processes are the dominant source of suspended sediment flowing from the Control gully. However, the Referee makes a good point that erosion processes in the catchment, possibly sourced from surface erosion, appear to be a major contributor of suspended sediment flowing through the gully systems and that the collection of accurate and representative catchment monitoring data is very important to understanding these dynamics. The authors will revise the existing commentary on catchment suspended sediment contribution, in the Results and Discussion and Conclusions sections, to provide emphasis on the importance of monitoring locations for the purpose of collecting representative catchment overland flow samples.

Note text has been added to the caption of Figure 1 regarding the placement of PASS samplers in the gully catchments.

*Line 138-140:* "N.B. The overland PASS sampler in sub-catchment 3 of the remediated gully was deployed several metres away from the flow line inferred by surface geology due to the redirection of flow associated with vegetation and termite mounds."

**Specific comment 3.** "Also, how do the catchment areas compare? The total catchment area of the remediated gully is ~13ha but what is the catchment area above each PASS sensor in the sub-catchments and how does this play into the results? And the catchment area for each sub-catchment of the remediated gully."

*RESPONSE:* ***Acknowledge.*** The aim of measuring suspended sediment in water flowing overland into the gullies was to understand their contribution, in terms of suspended sediment concentration and particle size distribution, to the suspended sediment measured at the gully outlet. Because of this, the size of the catchment is less important as ensuring that the major catchment drainage inputs into the gully are monitored. For example, the remediated gully catchment drains into the gully from three separate locations, later mixing at a confluence within the gully. Thus, three monitoring locations were required to account for the majority of overland flows draining from the catchment. In contrast, the majority of catchment overland flows into the control gully drain through one location, thus, it was monitored at one location up-stream of the gully head. Evaluation of the influence of sub-catchment area on the contribution of suspended sediment to the gullies would require the estimation of suspended sediment loads from these sub-catchments. As discussed in the manuscript (Lines 290-295 and 463-466) the estimation of loads from these highly ephemeral systems, and their catchments, is very challenging and was not feasible for this study. The authors acknowledge the importance of the catchment area regarding overland flow sediment contributions and will provide commentary on this in the Results and Discussion section.

Commentary regarding this was added to the Results and Discussion:

***Line 403-406:*** "In contrast, the PSD of suspended sediment samples collected from the outlet of the remediated gully (d50 of 5.84 μm) and samples collected from Catchment 2 (d50 of 5.52 μm) and 3 (d50 of 5.06 μm) of the three catchment areas draining into the gully were very similar, thus suggesting their sediment contributions would be similar, when normalised for differences in catchment area size."

**Specific comment 4 .** "If possible, I think a before remediation and after remediation DEM image (or DEM of difference maybe) would be a useful addition."

*RESPONSE:* ***Accept.*** Please refer to Figure 2.

**Specific comment 5 .** "Figure 7 and 8 seem to suggest that the fine fraction is coming from the catchments more so than the gullies? But there isn't much discussion about this? Maybe I'm interpreting the results wrong but if this is the case, I think it's one of the more interesting findings for discussion."

> *RESPONSE:*      *Clarify.*   Figure 7 and 8 demonstrate that the distribution of fine sediment (0.1-30 μm) in the suspended sediment samples collected from the catchments and gully outlets are similar. However, further investigation (e.g., geochemical tracing) would be required to differentiate the sources of fine sediment in the gully outlet sample. The authors agree with the referee's observation that the catchments samples appear to consist of mostly fine sediment as is indicated by in Figures 7 and 8 and Table 3 ($d_{90} < 36$ μm for all overland flow samples collected). Given this it could be suggested that the catchments contribute some of the fine suspended sediment measured at the gully outlet.
>
> This is discussed in Section 3.2.3 Particle size distribution, Lines 356-368: "The suspended sediment PSD characteristics of control gully catchment PASS samples was notably different to the gully outlet PASS samples (Table 3). This indicates the contribution of slightly coarser suspended sediment from gully erosion (d50 10.8 μm) is greater than the suspended sediment contribution of the catchment (d50 4.29 μm) in the control gully. In contrast, the PSD characteristics of suspended sediment samples collected from the outlet of the remediated gully (d50 of 5.84 μm) and samples collected from Catchments 2 (d50 of 5.52 μm) and 3 (d50 of 5.06 μm) of the three catchment areas draining into the gully were very similar (Table 3) (Figure 8). This suggests there is a notable contribution of sediment entering both gullies from their respective catchments.".

**Specific comment 6.** "In your abstract and conclusions you present a value of 80% as the sediment reduction achieved but it's not clear how this number is calculated? Is it the (SSC control – SSC remediated) / (SSC control)? Or some other number?"

> *RESPONSE:*      *Clarify.* The sections the Referee mentions, state the following: Lines 22-23: "Suspended sediment concentrations were ~80% lower at the remediated site compared to the control site,…" and Lines 460-463: "The multiple lines of evidence from this water quality study indicate the application of intensive landscape-scale remediation on actively eroding alluvial gullies has the potential to reduce average suspended sediment concentrations by more than 80%." These statements imply that the SSCs of the different gullies were compared and the difference in concentration between the two was ~80%. This comparison is discussed in further detail in Section 3.2.1 Suspended sediment concentration. It is not uncommon to see statements such as these without detailed explanations of the exact formula used in the abstract conclusion sections of a scientific journal article.

**Specific comment 7.** "[Line] 52: "There are various types of gullies present in the GBR catchment region (e.g., hillslope, colluvial, ephemeral, and soft-rock badlands), however, alluvial gullies likely represent the largest source of sediment, accelerated by land use change, to the GBR." - Reference?"

> *RESPONSE:*      *Accept.* The text now states:
>
> ***Line 63-65:*** "Of the various types of gullies present in the GBR catchment region (i.e., hillslope, colluvial, ephemeral, and soft-rock badlands), alluvial gullies likely represent the largest source of sediment to the GBR (Brooks et al., 2013; Brooks et al., 2020a)"

**Specific comment 8.** " [Line] "90: "The study site topography is relatively flat." - Would be good to know average slope?"

> *RESPONSE:*      ***Accept.*** The following commentary has been added:
>
> ***Line 112-113***: "The study site topography is relatively flat (average slope of the gullies and respective catchments at the site ranges from 8.6 – 9.7 m/m) with undulating gradients, surrounded by sandstone ranges."

**Specific comment 9.** " 101: "Erosion rates derived from repeated airborne LiDAR between 2009 and 2015 (before remediation activity), indicate the control gully produced slightly more sediment (61 t -1 ha-1 yr-1 ) compared to the remediated gully (50 t-1 ha-1 yr-1 ), based on gully catchment area." - Per unit area of gully or catchment?"

> *RESPONSE:*      ***Clarify.*** The authors reference the sediment yields estimated by Brooks et al., 2016, where the unit area was inclusive of the gully and its associated catchment area.

**Specific comment 10.** [Line] **"102 – 103: t-1 ha-1 yr-1 » t . ha-1 . yr-1** mass shouldn't be a reciprocal here.

> *RESPONSE:*      ***Accept.*** The text now states:
>
> ***Line 125-128***: "Erosion rates derived from repeated airborne LiDAR collected before remediation was conducted (2009 to 2015), indicate the control gully produced slightly more sediment (61 t/ha/yr) compared to the remediated gully (50 t/ha/yr), based on gully catchment area (Brooks et al., 2016)."

**Specific comment 11.** [Line] "103: "Note, LiDAR does not account for the surface erosion generated from the catchment area of each gully, which would be expected to be comparable on an area normalised basis. Hence, the difference in specific yields between the treatment and control would be less than indicated by the LiDAR data alone (Brooks et al., 2016)." - I find this statement a little confusing. I think you either need to be clearer about what this means or not include it.

> *RESPONSE:*      ***Accept.*** The authors decided to remove the statement.

**Specific comment 12.** [Line 169]: "time weighted average (TWA) SSC" – I can take a guess at what this is but it would be nice to have an equation.

> *RESPONSE:*      ***Accept.*** The following text and equation have been added:
>
> ***Line 224-230:*** TWA SSC of PASS samples was determined using equation 1:
>
> $$TWA\ SSC\ (mg/L) = \frac{M}{tF}$$
>
> Where the total mass of suspended sediment collected by the sampler ($M$; milligrams) is divided by the volume of water sampled during deployment (duration of sampler operation ($t$; minutes) multiplied by the pump flow rate ($F$; litres per minute))."

**COMMENTS FROM REFEREE 3:**

**GENERAL COMMENTS FROM REFEREE 3:**

**General Comments 1.** "GENERAL REMARKS The reviewed manuscript refers to the interesting topic on remediation measures used to decrease the negative impact of gully erosion. Such studies are highly needed, especially when they are carried out in one of the most valuable area around the world as the Great Barrier Reef. I appreciate that Authors tested different monitoring methods and evaluated them. These findings may be useful in other areas characterized by dispersive soils and intense short rainfall events. In my opinion this manuscript fits to the scope of Hydrology and Earth System Sciences journal. The methods are clearly presented (some minor remarks are marked below). The results and conclusions are generally clear, concise, and well-structured. Although, I think that this section can be improved. It would be great to see some comparison of remediation measures used in this study with studies from other regions. The figures are readable, and they correspond well with the data presented in supplement. In order to improve the quality of the paper, I include below some minor remarks."

> *RESPONSE:* **Acknowledge**. The authors acknowledge this positive comment and have undertaken specific reconsideration to address the other key points raised by the reviewer (see below).

**SPECIFIC COMMENTS FROM REFEREE 3:**

**Specific Comment 1.** " Lines 1-4 Please, consider shortening the title."

> *RESPONSE:* **Accept.** The revised title is "Intensive landscape-scale remediation improves water quality of an alluvial gully located in a Great Barrier Reef catchment".

**Specific Comment 2.** "Lines 20-21 I suggest to include some information on methods to the abstract. Now you just wrote that novel monitoring network was used without any details."

> *RESPONSE:* **Accept.** The following statement was added to the abstract:
>
> ***Line 20-24:*** "The gully remediation method was a first attempt, in the region, to invest a high level of financial (total cost of remediation ~$90,000 AUD) and logistical effort (e.g., intensive earthworks including the establishment of an on-site quarry) to develop long lasting erosion mitigation measures (i.e., regraded, compacted and battered gully walls, rock armouring of banks and channel, and installation of rock check dams)."

**Specific Comment 3.** "Can you refer also to the studies on remediation measures in other areas, not only in the GBR catchments?"

> *RESPONSE:* **Accept.** Text has been added to the introduction that states:
>
> ***Line 71-79:*** " A recent review by Bartely and co-workers (2020) identified several scientific studies that evaluated the effectiveness of gully remediation on improving water quality in various regions around the world, including: the French Alps (Mathys et al., 2003), Southern regions of the United States of America (Polyakov et al., 2014; Nichols et al., 2016), Spain (Hevia et al., 2014), China (Rustomji et al., 2008; Wang et al 2011), and Ethiopia (Ayele et al., 2018; Dagnew et al., 2015). Bartely and co-workers concluded that remediation efforts generally decrease the sediment yield of eroding gullies and thus improve water quality conditions. However, water quality improvements were driven by the extent of remediation (catchment and gully) and the re-establishment of vegetation in the gully post-remediation (Bartely et al., 2020)."

**Specific Comment 4.** "Line 54 Slacking or slaking?"

> *RESPONSE:* ***Clarify.*** The authors thank the Referee for pointing out this oversight. We believe the correct term is slaking sediments. This spelling mistake has been corrected.

**Specific Comment 5.** "Line 95 I'm confused. You wrote in the text that you used two gullies in the study, whereas in Figure 1 you marked three remediated gully catchments and one control gully catchment. Were these three gully catchments treated as one? Can you mark them together for instance with the same colour line or somehow marked them as one site?"

> *RESPONSE:* ***Clarify/Accept.*** Please refer to the revised Figure 1.

**Specific Comment 6.** "Lines 120-129 I suggest to include some photos from the study area. I know that you present several photos in the supplement, but I think that some of them should be in the manuscript, e.g.., control gully, remediated gully before and after remediation."

> *RESPONSE:* ***Accept.*** Please refer to Figure 2.

**Specific Comment 7.** "Lines 187-192 Did you analyse the whole soil profiles or did you only take samples from the topsoil/subsoils? At which depth did you take samples? Why did you put this subsection (2.4.3. Soil sampling and analysis) into section 2.4. Monitoring methods? I suppose that you did these analyses only once and PSD in soils wasn't monitored."

> *RESPONSE:* ***Acknowledge/Clarify.*** Whole soil samples were analysed for particle size distribution using hydrometer techniques. Soil samples were collected from the face the gully (i.e., the areas undergoing erosion) at depths ranging from the surface to 1 m. The soil sampling and analysis section was written as a separate section because these analyses were only conducted once and the authors thought it best not to group it under the water quality monitoring methods section.
>
> The text now states:
>
> ***Line 209-217:*** "Soil samples were collected as part of the design phase of the gully remediation project (Brooks et al., 2016; Brooks et al., 2018). Soil samples (1-2 kg) were collected from the face and walls of the gullies (i.e., the areas undergoing erosion) using a hand trowel and auger at depths ranging from the surface to 1 m. 21 and 9 samples were collected from the remediated and control gullies, respectively, prior to the remediation activities. The soil samples were analysed for particle size distribution using the soil hydrometer method (ASTM standard method 152H) (Brooks et al., 2016). Soil particle size distribution data was composited and treated as an average for the purpose of comparing gully soil to suspended sediment. This was done as soil to 1m deep can be eroded into suspended sediment during a flow event (e.g., gully wall collapse can impact large sections of the headscarp and expose deeper erodible soils) (Garzon-Garcia et al., 2016)."

**Specific Comment 8.** "Line 194 Which samples? I suppose that suspended sediments, but it should be clarified."

> *RESPONSE:* ***Accept/clarify.*** The authors thank the Referee for pointing out this oversight. The sentence now states the following in the revised manuscript:
>
> ***Line 221-223***: "Water samples collected from the Remediated and Control gullies were analysed for suspended sediment concentration (ASTM standard method D 3977-97), and particle size distribution using laser diffraction spectroscopy (Malvern Mastersizer 3000, Malvern Instruments)."

**COMMENTS FROM PETER HAIRSINE:**

**GENERAL COMMENTS FROM PETER HAIRSINE:**

**General Comments 1.** "This manuscript addresses a clear knowledge gap: the evaluation of a remediation technique for alluvial gullies in a tropical setting."

> *RESPONSE:* ***Acknowledge***. The authors acknowledge this positive comment and have undertaken specific reconsideration to address the other key points raised by Peter Hairsine (see below).

**SPECIFIC COMMENTS FROM PETER HAIRSINE:**

**Specific Comment 1.** "The primary conclusion is given as "The multiple lines of evidence from this water quality study indicate the application of intensive landscape-scale remediation on actively eroding alluvial gullies has the potential to reduce average suspended sediment concentrations by more than 80%." This conclusion is made on the basis of a comparison of 2 wet seasons of suspended sediment measurement using 4 different types of water samplers for a single control (3.3 hectares catchment area) and single treated catchment (13.7 hectares catchment area) - noting that 3 treatment catchment areas are labelled in figure 1."

The recent review of gully remediation efficacy of Bartley et al. (2020) demonstrates other similar studies have combined multi-year monitoring, pre-treatment measurement and replication in a range of settings. No study was found to provide a "gold standard" BACI, multi-decadal and replicated study but conclusions and attribution were normal reduced as a result."

> *RESPONSE:* ***Clarify.*** The authors thank Peter for notifying the authors of the recent study completed by his colleagues (Bartely et al., 2020). This manuscript was submitted to HESS prior to the publication of the Review article Peter mentions. The authors have included the findings of the review in the revised manuscript literature review.
>
> ***Line 72-79:*** "A recent review by Bartely and co-workers (2020) identified several scientific studies that evaluated the effectiveness of gully remediation on improving water quality in various regions around the world, including: the French Alps (Mathys et al., 2003), Southern regions of the United States of America (Polyakov et al., 2014; Nichols et al., 2016), Spain (Hevia et al., 2014), China (Rustomji et al., 2008; Wang et al 2011), and Ethiopia (Ayele et al., 2018; Dagnew et al., 2015). Bartely and co-workers concluded that remediation efforts generally decrease the sediment yield of eroding gullies and thus improve water quality conditions. However, water quality improvements were driven by the extent of remediation (catchment and gully) and the re-establishment of vegetation in the gully post-remediation (Bartely et al., 2020)."

**Specific Comment 2.** "I looked for "multiple lines of evidence" but only found the suspended sediment sampling with 4 devices arranged side by side."

> *RESPONSE:* ***Acknowledge/Clarify.*** For context, Peter is referring to Line 460 "The multiple lines of evidence from this water quality study indicate the application of intensive landscape-scale remediation on actively eroding alluvial gullies has the potential to reduce average suspended sediment concentrations by more than 80%."
>
> In this statement authors are referring to the collection of water quality data using different monitoring methods (i.e., four different methods for collecting water quality data) to assess the effect of gully remediation on water quality. Each of the water quality monitoring methods used collect a sample in a manner that is independent compared to the others, thus, evidence is provided by four separate lines of data gathering. Furthermore, the use of different water quality analyses provides further relevant lines of evidence, that are complimentary to the separate collection methods, (i.e., suspended sediment concentration, particle size distribution, and nutrient and carbon analyses).
>
> The authors acknowledge that the term "multiple lines of evidence" may mislead some readers to thinking that complimentary data (i.e., Lidar soil loss estimates) are referenced here. Thus, the authors have revised the text to state:
>
> ***Line 505-507:*** "The water quality data collected during this study, using multiple monitoring methods, supports the application of intensive landscape-scale remediation to significantly reduce suspended sediment concentrations in actively eroding gullies."

**Specific Comment 3.** "The two years of sampling does not enable any assessment of whether the hydrological forcing can be interpreted in terms of long term rainfall variability. Furthermore this sampling does not necessarily represent the long term (decades) performance of the remediation measures."

> *RESPONSE:* ***Acknowledge.*** The authors agree with Peter in that there is a need for more monitoring data, over longer time scales, to evaluate the effects of long term stressors (i.e., rainfall variability and backwater flooding effects). The authors infer this sentiment in the final statement of the conclusion section Line 477: " However, more information is needed, particularly sediment load estimates and information on remediation longevity over decadal timescales."
>
> The following statements have been added to the results and discussion and conclusion sections:
>
> ***Line 319-321:*** "However, because this study only includes two wet seasons of data it should be considered preliminary until it is further validated by continued monitoring of the remediated gully for several additional wet seasons."
>
> ***Line 506-509:*** "Further monitoring at the site should be conducted over longer timescales (i.e., decades) to evaluate the longevity of the erosion mitigation controls used as part of the gully remediation works."

**Specific Comment 4.** "The difference in performance between the control and treatment gullies is well summarised in terms of the suspended sediment concentrations and the particle size distributions. The difference between these measures is then attributed to the treatment effect. While this step is intuitive, it is not formally supported given the many limitations of the methodology as noted above."

> *RESPONSE:* ***Clarify.*** The aim of the study was to determine if gully water quality conditions were improved by landscape-scale remediation using water quality monitoring methods that employ sampling processes that are unique from one another. This was done so that limitations of one or more of the monitoring methods used could be accounted for when evaluating the effectiveness of the remediation measures. The water quality of the overland flow waters draining into the Remediated and Control gullies from their respective catchments is relatively similar. The overland flow water represents the major transport mechanism for suspended sediment within the gully system. Thus, the lower concentrations of suspended sediment and associated nutrients and carbon in the Remediated gully compared to the Control gully can only be attributed to the reduction in sediment and nutrient sources (i.e., erodible soil) from within the gully itself. This is not an intuitive assessment, rather, it is an interpretation of the data gathered. Furthermore, the authors have provided before and after photographs and digital elevation imagery of the remediated gully that demonstrates how erosion of the gully system has been greatly reduced, since remediation. This complimentary line of evidence will further support the conclusions made regarding the effectiveness of the remediation measures used at the Remediated gully.

[revised manuscript text omitted]

---

## Author Response (AR2)

**COMMENTS FROM REFEREE 1:**

**GENERAL COMMENTS FROM REFEREE 1:**

**General Comments.** The authors have answered all the comments and suggestions provided by all the reviewers. I consider that the manuscript has been improved and that it can be published after some minor revisions.

> *RESPONSE:* ***Acknowledge****.* The authors acknowledge this positive comment and have undertaken specific reconsideration to address the other key points raised by the referee (see below).

**SPECIFIC COMMENTS FROM REFEREE 1:**

**Specific Comment 1.** " My main point is still the limitation of the study period and the limitation of the measurements (the same problem has been also identified by the different reviewers). the authors have noted this problem in the manuscript, adding that it is preliminary information and that further monitoring program should be carried out. This stage, my main point is that this issue should be highlighted also in the abstract and conclusion sections."

> *RESPONSE:* ***Accept****.* The authors accept the referees concern regarding limitation of two years monitoring being relatively short. This limitation is now included in the abstract and conclusion sections:
>
> **Line 30-33:** "Monitoring was conducted during two consecutive wet seasons and thus can only provide preliminary information. Monitoring over longer time scales (i.e., 5-10 years) will need to be carried out in-order to validate the findings discussed herein."
>
> **Line 513-516:** "The findings from this study regarding the longevity of the erosion mitigation controls used as part of the gully remediation works are considered to be preliminary, pending the results of monitoring data collected from the site over longer timescales (i.e., semi-decadal to decadal)."

**Specific Comment 2.** "In my previous review, I indicate that an initial research hypothesis should be included. Please check it in the new text."

> *RESPONSE:* ***Accept*** The authors have included an initial research hypothesis in the introduction.
>
> **Line 101-103:** "We hypothesise that the application of landscape-scale gully erosion control measures (i.e., gully reshaping, soil compaction, rock armouring of channels and banks, and the installation of check dams) will cause a reduction in suspended sediment and nutrient concentrations at the study site."

**Specific Comment 3.** "Line 44. Check the spelling of Poesen."

> *RESPONSE:* ***Accept*** The authors thank the Referee for noticing the misspelled citation. The mistake has been corrected.
>
> **Line 46:** "…(Poesen et al., 2011;…